# The Clinical Characteristics and Treatments for Large Cell Carcinoma Patients Older than 65 Years Old: A Population-Based Study

**DOI:** 10.3390/cancers14215231

**Published:** 2022-10-25

**Authors:** Anjie Yao, Long Liang, Hanyu Rao, Yilun Shen, Changhui Wang, Shuanshuan Xie

**Affiliations:** 1Department of Respiratory Medicine, Shanghai Tenth People’s Hospital, School of Medicine, Tongji University, Shanghai 200072, China; 2School of Medicine, Tongji University, Shanghai 200092, China; 3Shanghai Songjiang District Jiuting Town Community Healthcare Center, Shanghai 201600, China

**Keywords:** large cell carcinoma, non-small cell lung cancer, radiation after surgery, multivariate cox analysis, overall survival

## Abstract

**Simple Summary:**

Pulmonary large cell carcinoma (LCC) represents a kind of rare and highly malignant tumors with significantly worse survival outcomes compared to other types of NSCLC. Our study mainly demonstrated that for LCC patients ≥65 years old, radiation after surgery had the optimal therapeutic effect to improve survival outcomes compared to other sequences of radiation with surgery. Our research provided significant advice on the appropriate choice of radiation sequences with surgery for advanced LCC patients with age ≥65 years old.

**Abstract:**

Background: Pulmonary large cell carcinoma, a type of non-small cell lung cancer (NSCLC), is a rare neoplasm with poor prognosis. In this study, our aim was to investigate the impact of radiation sequences with surgery for stage III/IV LCC patients between different age groups, especially in the elderly patients. Patients and Methods: The patients with LCC and other types of NSCLC in the Surveillance, Epidemiology and End Results (SEER) database from 2004 to 2015 were retrospectively analyzed. Then we divided the LCC patients into two age groups: <65 years old group and ≥65 years old group. Propensity score method (PSM) was used to control potential differences between different groups. The overall survival (OS) of LCC patients and other types of NSCLC patients were evaluated by Kaplan–Meier analysis. Univariate and multivariate Cox regression analysis were employed to explore the independent risk factors of OS. The forest plots of HRs for OS were generated to show the above outcomes more visually. Results: In total, 11,349 LCC patients and 129,118 other types of NSCLC patients were enrolled in this study. We divided LCC patients into <65 years old group (4300) and ≥65 years old group (7049). LCC patients was more common in whites (81.4%), males (58.3%), elderly (≥65 years old: 62.1%), east regions (52.7%), upper lobe (51.6%), right-origin of primary (55.4%), with advanced grade (54.2%) or stage (76.7%). After PSM, Kaplan–Meier analysis and multivariate Cox analysis showed significantly worse survival prognosis for LCC patients compared to other types of NSCLC, especially in the group ≥65 years old (HR: 1.230; 95% CI: 1.171–1.291; *p* < 0.001). For LCC patients, there were some risk survival factors including whites, males, not upper lobe, advanced stage, elder age at diagnosis, bone metastasis, liver metastasis, singled status, no lymphadenectomy, no surgery, and no chemotherapy (*p* < 0.05). In LCC patients ≥65 years old, radiation after surgery had significantly better impact on overall survival outcomes (HR: 0.863, 95% CI: 0.765–0.973, *p* = 0.016), whereas radiation prior to surgery (HR: 1.425, 95% CI: 1.059–1.916, *p* = 0.019) had significantly worse impact on prognosis of patients. In LCC patients <65 years old, radiation sequences with surgery had no significant impact on the OS of patients (*p* = 0.580), but ≥4 LNRs had significantly survival benefits to prognosis (HR:0.707, 95% CI: 0.584–0.855). Elderly LCC patients had worse malignant tumors than young patients, of which the majority were diagnosed as stage III/IV tumors. Conclusions: Postoperative radiotherapy may achieve a better prognosis for stage III/IV LCC patients older than 65 years old compared to other radiation sequences with surgery.

## 1. Introduction

According to the American Cancer Society, lung cancer remains the leading cause of the cancer death [1]. Non-small cell lung cancer (NSCLC), the most common lung cancer (approximately 85% of lung cancer), is generally classified into 3 major types: adenocarcinoma (about 40%), squamous cell carcinoma (about 25% to 30%), and large cell carcinoma (about 5% to 10%) [2]. Previously, large cell carcinoma (LCC) was defined as lung cancer which lacked any morphologic differentiation of small cell carcinoma, glandular carcinoma or squamous carcinoma observed by hematoxylin-eosin (HE) staining method, which has subtypes including large cell neuroendocrine carcinoma, large cell carcinoma with rhabdoid phenotype and large cell carcinoma, basaloid carcinoma, lymphoepithelioma-like carcinoma and clear cell carcinoma [3]. But according to the 2015 World Health Organization classification of lung cancer, LCC was redefined as surgically resected tumors which lacked clear evidence of glandular, squamous, neuroendocrine or any other immunohistochemical differentiation based on the immunohistochemical (IHC) markers and genetic molecular testing, which made fewer tumors be diagnosed as LCC [4,5].

Pneumonectomy with lymphadenectomy was generally recommended as the standard surgery for early-stage NSCLC patients [6], whereas chemotherapy and radiotherapy were of benefit to the survival of advanced stage NSCLC patients with organ metastasis [7,8]. As a kind of rare and poorly differentiated NSCLC, the majority of IHC-null LCCs were advanced tumors such as stage III/IV tumors, and could lead to significantly worse patient survival outcomes than for those with other IHC-positive subtypes of NSCLC [9,10]. Therefore, LCC patients generally combined different treatments together, in which radiation with surgery was a kind of common therapeutic strategy, especially in the advanced stage patients. Preoperative radiotherapy could downstage tumors originally considered inoperable to perform the following surgery successfully, whereas postoperative radiotherapy could control possible pathologic disease left after surgery to reduce recurrence and metastasis. However, due to the risk of toxicities and complications of radiation, there has been debate on the appropriate clinical cases and exact survival benefits of preoperative and postoperative radiotherapy for a long time [11]. According to the national comprehensive cancer network (NCCN) guidelines, postoperative radiotherapy or preoperative radiotherapy was an option for NSCLC patients with locally advanced tumors, such as resectable stage IIIA pathologic N2 tumors, but the optimal time of radiotherapy with surgery was still controversial [12]. An expert consensus on Adjuvant Therapy of NSCLC from China Thoracic Surgery Committee proposed that postoperative radiotherapy could be considered for stage III-N2 NSCLC patients with lymph nodes metastasis, but was not recommended for stage I-II NSCLC patients [13]. 

Due to the rare incidence, high malignancy of LCC and a lack of clinical data, little is known about the patients’ biological and clinical characteristics and appropriate treatments, let alone the radiation sequences with surgery. In this retrospective study, we collected and analyzed the biological and clinical data on a large number of LCC patients registered in the Surveillance, Epidemiology and End Results (SEER) database, and divided them into different age groups (<65 years old group and ≥65 years old group), in order to explore how different factors and treatments, especially the radiation sequences with surgery, affect the survival of stage III/IV LCC patients. 

## 2. Patients and Methods

### 2.1. Data Source

We used the US National Cancer Institute’s surveillance, epidemiology, and end results (SEER) database. The database collects patients’ information including disease types, disease stage, histologic category, treatment strategy and survival time, and covers approximately 28% of the US population before 2015 [14]. In this database, personally identifying information is excluded and the data are available publicly.

### 2.2. Study Population

We limited the cohort to patients who were histologically diagnosed with epithelial tumors, including adenocarcinoma (pathological codes 8140/3), squamous cell carcinoma (pathological codes 8070/3), large cell carcinoma (pathological codes 8012/3), large cell neuroendocrine carcinoma (pathological codes 8013/3) and other types of NSCLC from 2004–2015 [11]. In our research, patients with stages from I to IV were covered, while those without any cancer-directed treatment were excluded. Another exclusion criterion was the incomplete information for these parameters: age, microscopic diagnostic confirmation, demographic data, cause of death, and survival time. The detailed screening flowchart is shown in Figure 1.

### 2.3. Data Elements

Baseline patient and tumor characteristics included the following covariates: race, sex, year of diagnosis, region, primary site-labeled, grade, stage, histology, laterality, lymph nodes removed (LNRs) count, radiation sequences with surgery, radiation record, chemotherapy record, tumor size, bone metastasis, brain metastasis, liver metastasis, lung metastasis, survival time, first malignant primary indicator, age at diagnosis, insurance status, marital status, high school education (%), median family income (US dollars, in tens). In addition, information on high school education and median family income were obtained from the census track where the patients reside.

### 2.4. Statistical Analysis

All data of patients were analyzed by SPSS (version 25.0, SPSS Inc., Chicago, IL, USA). The propensity score method (PSM) was used to control potential differences between the LCC group and other types of NSCLC group, as well as the <65 years old LCC group and ≥65 years old LCC group, in which 1:1 without replacement and nearest-neighbor matching method of 0.02 was used. The survival curves were created by using the Kaplan–Meier method in which Log-rank was used to test the significant differences. Overall survival (OS) was calculated from the date of diagnosis to death from any cause. The factors with significant differences were analyzed by univariate COX regression analysis, followed by multivariate Cox regression analysis, to test the independent potential predictors of survival. The forest plots were generated by GraphPad Prism (version 8.0, GraphPad Software Inc, San Diego, CA, USA). In all of the analyses, a bolded *p*-value of less than 0.05 represents a significant statistical difference.

## 3. Results

### 3.1. Baseline Cohort Characteristics

We identified 11,349 LCC patients and 129,118 patients with other types of NSCLC. Firstly, we compared the clinical, histological, sociodemographic and therapeutic characteristics between the two groups before PSM (Table 1). Similar to other types of NSCLC patients, the elderly (patients ≥65 years old: 62.2%), whites (81.4%), males (58.3%) and east regions (52.7%) were more common in LCC patients. The majority of LCC tumors were located in the upper lobe (51.6%), along with the right laterality (55.4%), and smaller than 1 cm (75.5%), as well as other types of NSCLC patients. Most of the LCC patients had insurance (49.8%), spouse (52.6%), relatively more high school education (52.7%) and higher median family income (50.6%). Compared to other types of NSCLC patients, LCC patients were more likely to be diagnosed at a younger age, with a higher grade and stage of tumor and a greater chance of organ metastasis. Moreover, the majority of LCC patients refused to have surgery (78.0%), lymphadenectomy (72.4%), radiotherapy (56.8%) or chemotherapy (52.3%). Most LCC patients had no radiation and/or surgery (87.7%), whereas 10.7% of the rest had radiation after surgery, followed by radiation prior to surgery (1.4%). Beam radiation (41.2%) was the primary mode of radiation for LCC patients receiving radiotherapy, and ≥4 LNRs (14.8%) was the main modality for those accepting lymphadenectomy. The incidence of organ metastasis from high to low was bone (6.7%), brain (5.9%), liver (5.4%), lung (4.9%). After PSM, those characteristics were well balanced (Appendix A).

### 3.2. Survival Outcomes

All of the patients were well matched with PSM, generating 11,349 pairs. After PSM, the survival courses showed that the median survival time was significantly worse in the LCC group than in other types of NSCLC group (7 months vs. 10 months; Logrank *p* < 0.001; Figure 2A). For the LCC group and other types of NSCLC group, the 1-, 3- and 5-year OS rates were 34.5% vs. 44.1%, 15.7% vs. 21.1%, and 11.2% vs. 14.9%, respectively. According to ages, we divided the LCC patients into two groups: <65 years old group and ≥65 years old group, which generated 4068 pairs after PSM. The survival courses also showed that the median survival time was worse in ≥65 years old group than in <65 years old group (8 months vs. 7 months; Logrank *p* < 0.001; Figure 2B). The 1-, 3- and 5-year OS rates were 35.1% vs. 38.9%, 15.3% vs. 18.6%, and 10.3% vs. 14.5% for ≥65 years old group and <65 years old group, respectively. The median survival time for patients with organ metastasis from high to low was brain (5 months), lung (4 months), bone (3 months), liver (3 months) in <65 years old group. For the <65 years old group, the 1-,3-, and 5-year survival rate were 20.0%, 6.5%, and 3.5% in the brain metastasis subgroup, 18.0%, 3.4%, and 0.0% in the lung metastasis subgroup, 14.4%, 3.5% and 1.2% in the bone metastasis subgroup, and 11.2%, 1.5% and 0.0% in the liver metastasis subgroup, respectively. The median survival time for patients with organ metastasis from high to low was lung (4 months), brain (4 months), bone (3 months), liver (3 months) in the ≥65 years old group. For ≥65 years old group, the 1-,3-, and 5-year survival rate were 18.4%, 3.4%, and 2.6% in the lung metastasis subgroup, 13.8%, 2.0%, and 0.0% in the brain metastasis subgroup, 11.3%, 0.8% and 0.0% in the bone metastasis subgroup, 13.6%, 0.0% and 0.0% in the liver metastasis subgroup, respectively (Table 2). For subgroups of LCC patients <65 years old, patients with ≥4 LNRs, patients receiving surgery or chemotherapy had better survival rates than those refusing lymphadenectomy, surgery or chemotherapy (Logrank *p* < 0.001; Figure 3A–C). Moreover, radiation prior to surgery gave better survival outcomes than other radiation sequences with surgery in the <65 years old group (Logrank *p* < 0.001; Figure 3D). However, no significant difference was observed in the radiation record subgroup <65 years old (Logrank *p* = 0.418; Figure 3E). For subgroups of LCC patients ≥65 years old, patients with ≥4 LNRs, surgery or chemotherapy had better survival than those refusing lymphadenectomy, surgery or chemotherapy (Logrank *p* < 0.001; Figure 4A–C). However, radiation prior to surgery had no better survival outcomes than other radiation sequences with surgery (Logrank *p* < 0.001; Figure 4D). No significant difference was observed in the radiation record subgroup ≥65 years old (Logrank *p* = 0.509; Figure 4E).

### 3.3. Univariate and Multivariate Analysis

We analyzed different groups by univariate and multivariate COX regression analysis successively according to histology and age in all of the NSCLC patients enrolled and LCC patients respectively, then compared the different factors affecting the survival outcomes between different groups. For LCC patients, the multivariate COX regression analysis showed the following factors were related to survival risk including whites, males, not upper lobe, advanced stage, elder age at diagnosis, bone metastasis, liver metastasis, single status, no lymphadenectomy, no surgery, and no chemotherapy (*p* < 0.05, Table 3). Among these factors, the ≥65 years old group had significantly worse prognosis than the <65 years old group (HR: 1.230, 95% CI: 1.171–1.291, *p* < 0.001; Table 3). Therefore, we divided LCC patients into two groups: <65 years old group and ≥65 years old group. For the two different age groups, chemotherapy or surgery were of benefit to a survival prognosis (*p* < 0.001; Table 4 and Table 5). For LCC patients <65 years old, ≥4 LNRs status was also beneficial to survival of patients (HR: 0.707, 95% CI: 0.584–0.855; Table 4), whereas there were no significant survival differences observed in the radiation sequences with surgery subgroup (*p* = 0.580; Table 4). For LCC patients ≥65 years old, radiation after surgery (HR: 0.863, 95% CI: 0.765–0.973, *p* = 0.016) was of benefit to prognosis, whereas radiation prior to surgery (HR: 1.425, 95% CI: 1.059–1.916, *p* = 0.019) was harmful to survival outcomes (Table 5). In addition, the forest plots of HRs for OS were generated to show the same COX regression analysis outcomes of treatments between different age groups more visually (Figure 5). 

## 4. Discussion

In this study, we mainly explored the impact of clinical characteristics and therapeutic strategies on the survival outcomes in LCC patients, especially the elderly patients. The following factors were related with a higher risk of death in LCC patients: whites, males, not upper lobe, advanced stage, elder age at diagnosis, bone metastasis, liver metastasis, single status. There was a significantly worse survival prognosis in the ≥65 years old group who accounted for more than 60% of the LCC patients, than in the <65 years old group. Furthermore, surgery, lymphadenectomy, radiation or chemotherapy were all of benefit to the survival of LCC patients no matter whether young or aged. For LCC patients <65 years old, radiation sequences with surgery had no significant survival impact on survival time, but ≥4 LNRs had significantly survival benefits to prognosis of patients. However, for LCC patients ≥65 years old, radiation sequences with surgery had significant impact on overall survival. To be specific, radiation after surgery was the optimal radiation sequence with surgery. In addition, we found LCC patients were more likely to be whites, males and elders, which may be because of smoking [15]. Consistent with prior reports, LCC tumors were more commonly located in the upper lobe of the lung, along with right laterality [16]. Compared to other types of NSCLC patients, LCC patients were more likely to be diagnosed at a younger age, with a higher stage of tumors including stage III/IV tumors, and significantly worse survival outcomes; this was also consistent with other studies [17]. For all LCC patients, the incidence of metastatic organ from high to low was bone, brain, liver, lung. For LCC patients ≥65 years old, the 3-year of OS of metastatic organ from high to low was lung, brain, bone, liver. It was also reported that distant metastases to the tonsil, or gastrointestinal tract were quite rare with poor prognosis [18,19]. 

The standard treatment of LCC patients, especially the elder patients, has been debated for a long time because of the significant heterogeneity in the pathology and prognosis. Due to the rare incidence and poor prognosis of LCC, there were few reports focused on the survival effects of radiotherapy for LCC patients, not to mention the specific radiation sequences with surgery. We found that most of the LCC patients were diagnosed as advanced stage tumors such as stage III/IV LCC, and radiation after surgery had significant survival benefits to the LCC patients. A retrospective study enrolled 3197 LCC patients and demonstrated that radiotherapy combined with surgery may have a bad impact on survival for the stage I–III LCC patients, whereas radiation with surgery (HR: 0.394, 95% CI: 0.245–0.633, *p* < 0.001) could have a better survival impact than radiotherapy (HR: 0.767, 95% CI: 0.658–0.895, *p* < 0.001) or surgery (HR: 0.462, 95% CI: 0.297–0.720, *p* < 0.001) alone for the stage IV LCC patients, which was consistent with some of our views [20]. Another prospective study analyzed 4 stage III NSCLC patients after triple plastic resections, and found long-term survival benefits in a clinical case of a locally advanced LCC patient receiving postoperative radiotherapy, which indicated that postoperative radiation could be recommended for some advanced stage LCC patients with strict indications [21]. Some researches explored whether postoperative radiotherapy of NSCLC patients could also be valuable for the therapy of LCC patients because LCC is also a type of NSCLC. Several studies enrolled many resected stage III NSCLC patients and demonstrated that radiation after surgery could remarkably improve OS and reduced local recurrence, especially in the multiple-station pN2 group [22,23,24,25]. Recently a lung ART trial showed that conformal postoperative radiotherapy could improve disease-free survival (control group vs. experimental group: 44% vs. 47%) and local relapse-free survival (control group vs. experimental group: 46% vs. 25%), but was associated with an increased death rate (control group vs. experimental group: 5% vs. 15%) related to cardiopulmonary toxicities [26]. 

It is important to explore the appropriate therapy for the elder patients because the majority of the LCC patients were elders who were mainly diagnosed as stage III/IV tumors. Consist with some studies, we found the elderly patients were less likely to have received recommended surgery and/or radiotherapy in consideration of risk factors such as poor performance status, corresponding comorbidities, therapeutic complications and tolerance towards treatments [27,28,29]. However, we found that radiation after surgery had the optimal survival benefits to prognosis compared to other radiation sequences with surgery in the LCC patients ≥65 years old, but radiation with surgery had no survival effects in LCC patients <65 years old. Given that LCC is a type of poorly differentiated NSCLC, studies about postoperative radiotherapy in advanced NSCLC patients could also offer therapeutic references to the treatments of LCC patients to some extent. A propensity score-matching analysis enrolled 3334 resected stage IIIA-N2 NSCLC patients and demonstrated that postoperative radiotherapy could only offer significantly overall survival benefits to <60 years old group (5-year OS, 35.4% for postoperative radiotherapy vs. 28.9% for no postoperative radiotherapy; *p*  =  0.026), but not to 60–79 years old group (*p* = 0.062) and >80 years old group (*p* = 0.198) [30]. However, a retrospective study analyzed 17,654 stage IIIA N2 NSCLC patients and found that the surgery alone group did not have survival benefits compared to the no surgery group, and postoperative radiotherapy was recommended in patients >75 years old [31]. Another retrospective research based on 2515 stage IIIA N2 NSCLC patients showed that the survival benefits of delayed radiation after surgery was more significant in patients >60 years old (Logrank *p* = 0.002) compared to patients ≤60 years old (Logrank *p* = 0.871) [32]. This may because elder patients had more risk of lymph nodes metastasis than young patients, which needed radiation after surgery to prevent potential local recurrence. Therefore, elderly LCC patients need more accurate, more individual and more comprehensive therapeutic strategies in the future.

In addition, some LCC patients with advanced-stage tumors, especially the young patients, may not be very sensitive to chemotherapy or radiotherapy, so other comprehensive treatments based on surgical resection with lymphadenectomy are also an effective therapeutic strategy [33]. We recommended ≥ 4 LNRs as the optimal ways of lymphadenectomy in LCC patients <65 years old, but not in LCC patients ≥65 years old, which may because most of young patients had lower stage tumors, better health status and less surgical comorbidities than elder patients. Very few studies focused on the specific count of the LNRs in LCC, but many studies focusing on the NSCLC patients could also serve as references because LCC is also a type of NSCLC. A retrospective study found significant survival benefits for the incremental number of lymph nodes removed through to 4 lymph nodes in NSCLC patients [34], and another research found that the 5-year survivals of >6 LNRs group were better than that of ≤6 LNRs group in NSCLC patients according to the new lymph node descriptor proposed by the International Association for the Study of Lung Cancer (IASLC) [35]. Although the optimal number of LNRs remained controversial, a large number of studies agreed that a greater number of LNRs in a certain range of lymphadenectomy were related to more accurate lymph node staging and better long-term survival, which was consistent with our view [36]. Further, our research strongly recommended chemotherapy for LCC patients at any stage or age. Recently, there were several studies reporting that adjuvant chemotherapy, especially postoperative chemotherapy, had a significantly better prognosis than refusal of chemotherapy in LCC patients [37,38]. Considering the rare incidence and poor prognosis of LCC, it is significant to have regular health screening for high-risk individuals in order that they can be discovered, diagnosed and treated early. Recently, because of the additionally null histological and immunological definitions of LCC compared to other types of NSCLC, its diagnosis and treatment have undergone significant changes. More studies began to explore the immunotherapy and targeted therapy of LCC patients [39,40]. Hence, future research should be focused on how to combine radiotherapy with surgery or chemotherapy or immunotherapy or targeted therapy for LCC patients individually and comprehensively, in order to achieve long-term survival benefits.

The advantage of our study was that we had enrolled the largest number of LCC patient data so far according to the 2015 WHO classification. Moreover, the impact of different radiation sequences with surgery on the survival prognosis of LCC patients was discussed and the survival benefits of postoperative radiotherapy were demonstrated in our study for the first time. Most importantly, we compared many factors affecting survival of patients between different histology groups and age groups, and proposed that elderly LCC patients needed more personal and precise therapy. However, because of the lack of immunohistochemical information, the diagnosis of LCC was not accurate enough. The processing of clinical data may lead to a selection bias. There were also several other limitations in this study. The retrospective nature of study contributed to incomplete information on comorbidity score, performance status score and radiation dose and time. In addition, there were some new changes in materials and methods of radiotherapy over the recent years, which need to be explored in the future [41]. In a word, our research could still provide useful suggestions to the diagnosis and therapy of LCC patients in the future. 

## 5. Conclusions

In conclusion, LCC represents a kind of rare and highly malignant tumor with significantly worse survival outcomes compared to other types of NSCLC. Our study demonstrated that surgery, lymphadenectomy, chemotherapy or radiotherapy were all related to the improved OS in LCC patients. For LCC patients ≥65 years old, radiation after surgery had the optimal therapeutic effect to improve survival outcomes compared to other sequences of radiation with surgery. Our research provides significant advice on the appropriate choice of radiation sequences with surgery for advanced LCC patients with age ≥65 years old.

## Figures and Tables

**Figure 1 cancers-14-05231-f001:**
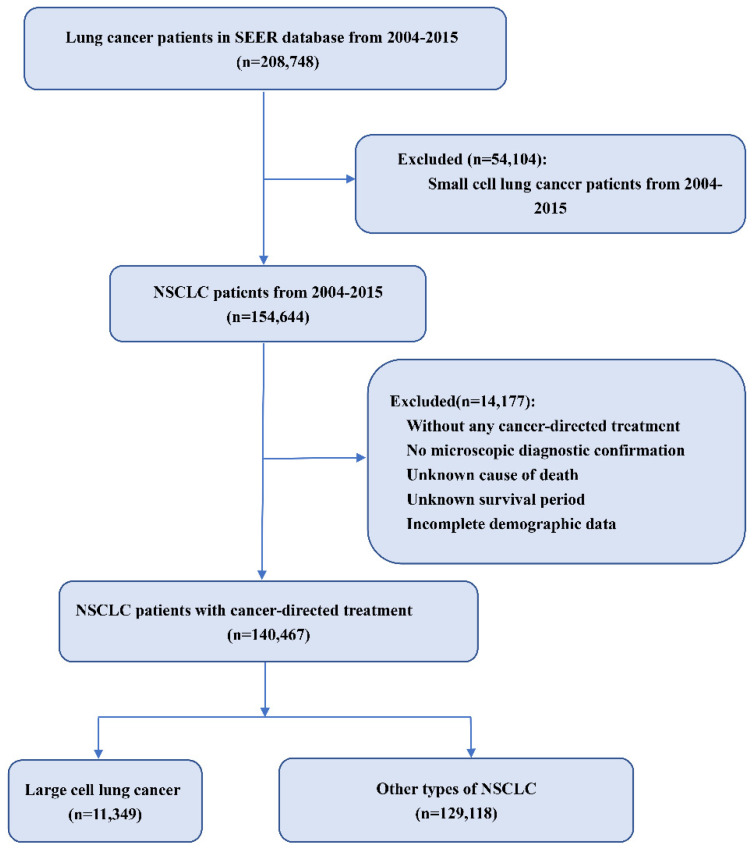
Flow chart of patient screening. Abbreviation: LCC: large cell carcinoma; NSCLC: non-small lung cancer.

**Figure 2 cancers-14-05231-f002:**
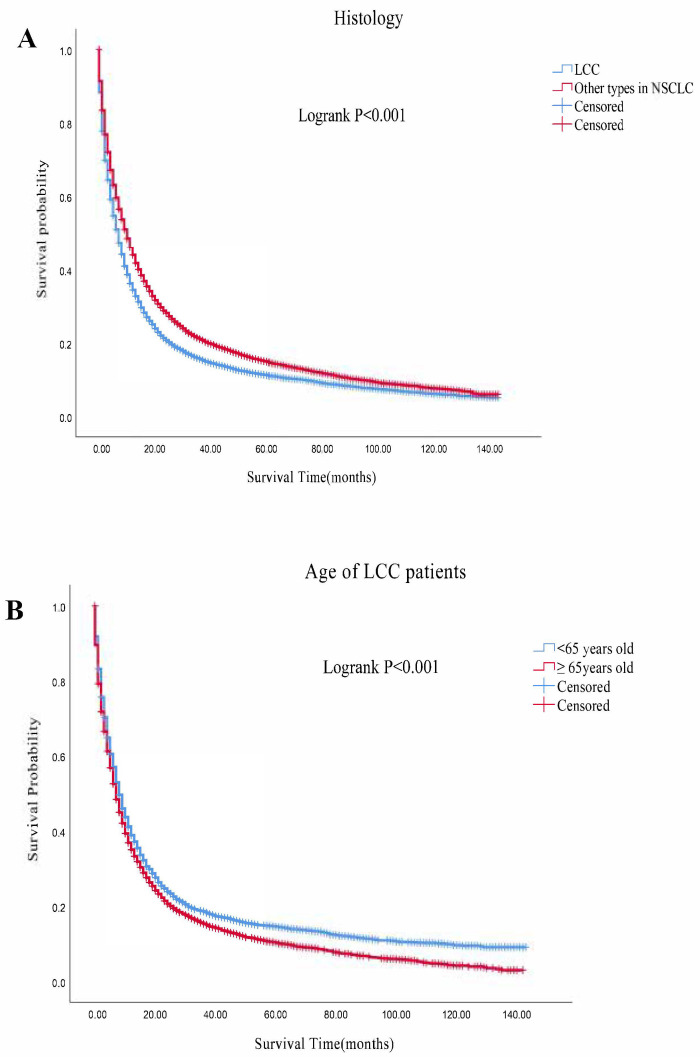
Comparison of survival curves of OS in LCC patients and other types of NSCLC patients (**A**) Comparison between LCC patients and other types of NSCLC patients; (**B**) Comparison between LCC patients <65 years old and ≥65 years old. Abbreviations: LCC: large cell carcinoma; NSCLC: non-small cell lung cancer; OS: overall survival.

**Figure 3 cancers-14-05231-f003:**
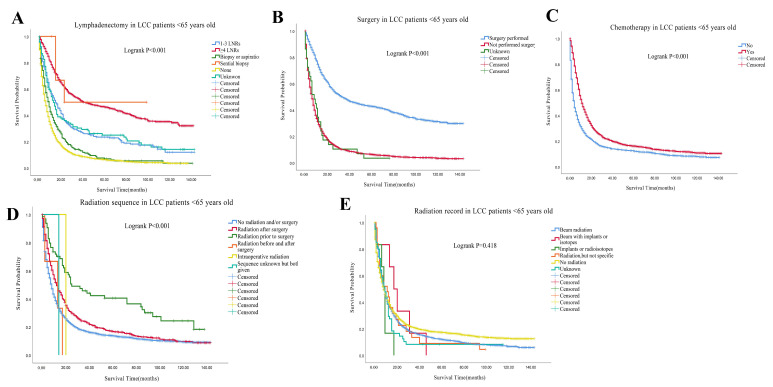
Comparison of survival curves of OS in different therapeutic subgroups of LCC patients <65 years old. (**A**) Comparison of lymphadenectomy subgroup; (**B**) Comparison of surgery subgroup; (**C**) Comparison of chemotherapy subgroup; (**D**) Comparison of radiation sequence subgroup; (**E**): Comparison of radiation record subgroup. Abbreviations: LCC: large cell carcinoma; LNRs: lymph nodes removed; OS: overall survival.

**Figure 4 cancers-14-05231-f004:**
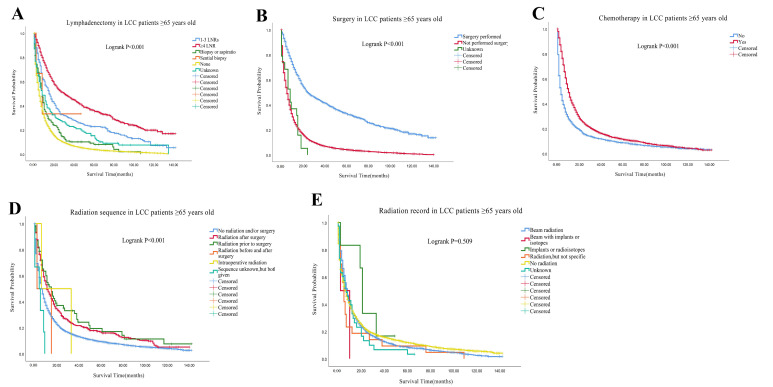
Comparison of survival curves of OS in different therapeutic subgroups of LCC patients ≥65 years old. (**A**) Comparison of lymphadenectomy subgroup; (**B**) Comparison of surgery subgroup; (**C**) Comparison of chemotherapy subgroup; (**D**) Comparison of radiation sequence subgroup; (**E**): Comparison of radiation record subgroup. Abbreviations: LCC: large cell carcinoma; LNRs: lymph nodes removed; OS: overall survival.

**Figure 5 cancers-14-05231-f005:**
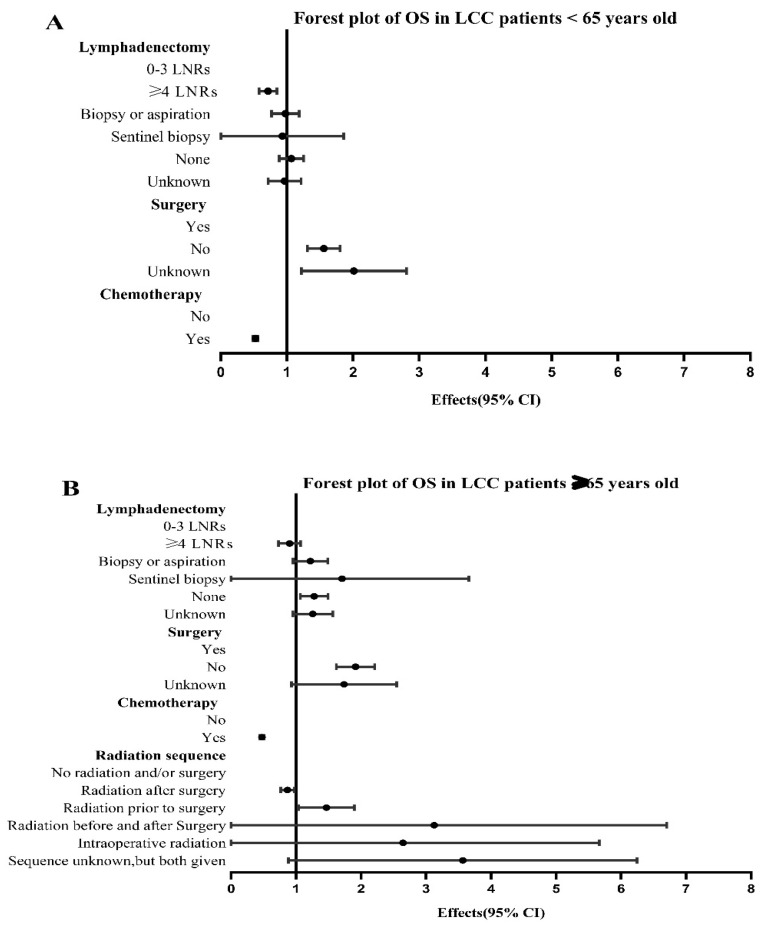
Forest plots of variables that can influence OS in LCC patients <65 years old and ≥65 years old. (**A**) Comparison of OS in LCC patients <65 years old; (**B**) Comparison of OS in LCC patients ≥65 years old. Abbreviations: LCC: large cell carcinoma; CI: confidence interval; OS: Overall survival; LNRs: lymph nodes removed.

**Table 1 cancers-14-05231-t001:** Baseline characteristics of patients with LCC and other types of NSCLC before PSM.

Characteristics	LCC (n = 11,349)	Others (n = 129,118)	*p*
Race			<0.001
White	9243(81.4%)	106,372(82.4%)	
Black	1568(13.8%)	15,398(11.9%)	
Asian and others	528(4.7%)	7170(5.6%)	
Unknown	10(0.1%)	178(0.1%)	
Sex			<0.001
Male	6618(58.3%)	78,075(60.5%)	
Female	4731(41.7%)	51,043(39.5%)	
Year of diagnosis			<0.001
2004–2007	5487(48.3%)	39,869(30.9%)	
2008–2011	3508(30.9%)	43,981(34.1%)	
2012–2015	2354(20.7%)	45,268(35.1%)	
Region			<0.001
East	5979(52.7%)	61,337(47.5%)	
Northern Plains	1237(10.9%)	14,743(11.4%)	
Southwest	343(3.0%)	3663(2.8%)	
Alaska and Pacific Coast	3790(33.4%)	49,375(38.2%)	
Tumor location			<0.001
Upper lobe	5857(51.6%)	65,624(50.8%)	
Middle lobe	465(4.1%)	4843(3.8%)	
Lower lobe	2573(22.7%)	37,529(29.1%)	
NOS	1761(15.5%)	11,856(9.2%)	
Overlapping lesion	137(1.2%)	1817(1.4%)	
Main bronchus	546(4.8%)	7147(5.5%)	
Trachea	10(0.1%)	302(0.2%)	
Grade			<0.001
Grade I	28(0.2%)	7401(5.7%)	
Grade II	95(0.8%)	33,817(26.2%)	
Grade III	3368(29.7%)	39,050(30.2%)	
Grade IV	2781(24.5%)	888(0.7%)	
Unknown	5077(44.7%)	47,961(37.1%)	
Stage			<0.001
Stage I	1269(11.2%)	23,033(17.8%)	
Stage II	668(5.9%)	10,475(8.1%)	
Stage III	2972(26.2%)	40,502(31.4%)	
Stage IV	5735(50.5%)	45,210(35.0%)	
Unknown	705(6.2%)	9898(7.7%)	
Laterality			<0.001
Right-origin of primary	6282(55.4%)	71,226(55.2%)	
Left—origin of primary	4420(38.9%)	53,577(41.5%)	
Bilateral, single primary	172(1.5%)	1401(1.1%)	
Paired, but no laterality	401(3.5%)	2184(1.7%)	
Others	74(0.7%)	730(0.6%)	
Lymphadenectomy			<0.001
0–3 LNRs	585(5.2%)	5949(4.6%)	
≥4 LNRs	1683(14.8%)	25,491(19.7%)	
Biopsy or aspiration	542(4.8%)	5829(4.5%)	
Sentinel biopsy	14(0.1%)	246(0.2%)	
None	8216(72.4%)	88,610(68.6%)	
Unknown	309(2.7%)	2993(2.3%)	
Surgery record			<0.001
Yes	2419(21.3%)	35,272(27.3%)	
No	8854(78.0%)	124,529(71.9%)	
Unknown	76(0.7%)	1062(0.8%)	
Radiation sequence			<0.001
No radiation and/or surgery	9951(87.7%)	115,975(89.8%)	
Radiation after surgery	1218(10.7%)	11,180(8.7%)	
Radiation prior to surgery	156(1.4%)	1611(1.2%)	
Radiation before and after surgery	12(0.1%)	197(0.2%)	
Intraoperative radiation	3(0.0%)	27(0.0%)	
Sequence unknown, but both given	14(0.1%)	88(0.1%)	
Surgery before and after radiation	0(0.0%)	28(0.0%)	
Radiation in and before/after surgery	0(0.0%)	12(0.0%)	
Radiation record			0.002
Beam radiation	4680(41.2%)	52,573(40.7%)	
Beam with implants or isotopes	8(0.1%)	204(0.2%)	
Implant or radioisotopes	18(0.2%)	269(0.2%)	
Radiation, but not specified	58(0.5%)	777(0.6%)	
No radiation	6445(56.8%)	74,095(57.4%)	
Unknown	140(1.2%)	1200(0.9%)	
Chemotherapy record			0.001
Yes	5408(47.7%)	57,335(44.4%)	
No/unknown	5941(52.3%)	71,783(55.6%)	
Tumor Size			<0.001
≤1 cm	8567(75.5%)	102,975(79.8%)	
>1, ≤2 cm	7(0.1%)	57(0.0%)	
>2, ≤3 cm	9(0.1%)	102(0.1%)	
>3, ≤4 cm	9(0.1%)	100(0.1%)	
>4 cm	6(0.1%)	62(0.0%)	
Unknown	2751(24.2%)	25,822(20.0%)	
Bone Metastasis			<0.001
Yes	759(6.7%)	8056(6.2%)	
No	2937(25.9%)	57,178(44.3%)	
Unknown	7653(67.4%)	63,884(49.5%)	
Brain Metastasis			<0.001
Yes	668(5.9%)	4139(3.2%)	
No	3024(26.6%)	61,029(47.3%)	
Unknown	7657(67.5%)	63,950(49.5%)	
Liver Metastasis			<0.001
Yes	610(5.4%)	4099(3.2%)	
No	3086(27.2%)	61,089(47.3%)	
Unknown	7653(67.4%)	63,930(49.5%)	
Lung Metastasis			<0.001
Yes	551(4.9%)	8226(6.4%)	
No	3125(27.5%)	56,758(44.0%)	
Unknown	7673(67.6%)	64,134(49.7%)	
First malignant primary indicator			<0.001
Yes	9213(81.2%)	98,320(76.1%)	
No	2136(18.8%)	30,798(23.9%)	
Age at diagnosis			<0.001
<65	4300(37.9%)	37,481(29.0%)	
≥65	7049(62.1%)	91,636(71.0%)	
Insurance status			<0.001
Any Medicaid	993(8.7%)	13,440(10.4%)	
Insured or no specifics	5655(49.8%)	81,576(63.2%)	
Uninsured	260(2.3%)	2372(1.8%)	
Unknown	4441(39.1%)	31,730(24.6%)	
Marital status			<0.001
Married or domestic partner	5975(52.6%)	66,147(51.2%)	
Divorced/separated/single/widowed	4995(44.0%)	57,548(44.6%)	
Unknown	379(3.3%)	5423(4.2%)	
High School education (%)			<0.001
≤10	2050(18.1%)	26,234(20.3%)	
>10, ≤20	5984(52.7%)	66,441(51.5%)	
>20, ≤30	2981(26.3%)	32,302(25.0%)	
>30	332(2.9%)	4128(3.2%)	
Unknown	2(0.0%)	13(0.0%)	
Median Family income (dollar, in tens)			<0.001
≤5000	1682(14.8%)	16,744(13.0%)	
>5000, ≤7000	5745(50.6%)	62,365(48.3%)	
>7000, ≤9000	2854(25.1%)	34,034(26.4%)	
>9000	1066(9.4%)	15,962(12.4%)	
Unknown	2(0.0%)	13(0.0%)	

Abbreviations: NSCLC: non-small cell lung cancer; LCC: large cell carcinoma. The *p*-value of less than 0.05 represents a significant statistical difference.

**Table 2 cancers-14-05231-t002:** Median survival months and 1,3,5-year of OS in LCC and other types of NSCLC patients.

	Median Survival Months	1-Year of OS (%)	3-Year of OS (%)	5-Year of OS (%)
Other types of NSCLC	10	44.1	21.1	14.9
LCC	7	34.5	15.7	11.2
LCC < 65 years old	8	38.9	18.6	14.5
Bone metastasis	3	14.4	3.5	1.2
Brain metastasis	5	20.0	6.5	3.5
Liver metastasis	3	11.2	1.5	0.0
Lung metastasis	4	18.0	3.4	0.0
LCC ≥ 65 years old	7	35.1	15.3	10.3
Bone metastasis	3	11.3	0.8	0.0
Brain metastasis	4	13.8	2.0	0.0
Liver metastasis	3	13.6	0.0	0.0
Lung metastasis	4	18.4	3.4	2.6

Abbreviations: LCC: large cell carcinoma; NSCLC: non-small cell lung cancer; LNRs: lymph nodes removed; NOS: not otherwise specified.

**Table 3 cancers-14-05231-t003:** Univariate and multivariate COX regression analysis on OS in LCC patients.

Variables	Univariate Analysis		Multivariate Analysis	
	HR (95% CI)	*p*	HR (95% CI)	*p*
Race		0.049		<0.001
White	Reference		Reference	
Black	1.072(1.005–1.144)	0.035	0.957(0.895–1.024)	0.203
Asian and others	0.913(0.819–1.019)	0.105	0.785(0.701–0.878)	<0.001
Unknown	0.935(0.389–2.247)	0.880	1.177(0.489–2.836)	0.716
Sex		<0.001		<0.001
Male	Reference		Reference	
Female	0.862(0.822–0.904)	<0.001	0.845(0.804–0.888)	<0.001
Year of diagnosis		0.760		
2004–2007	Reference			
2008–2011	0.980(0.929–1.034)	0.459		
2012–2015	0.992(0.929–1.060)	0.808		
Region		0.706		
East	Reference			
Northern Plains	1.036(0.959–1.119)	0.367		
Southwest	1.005(0.872–1.158)	0.946		
Alaska and Pacific Coast	0.986(0.935–1.040)	0.605		
Tumor location		<0.001		0.001
Upper lobe	Reference		Reference	
Middle lobe	1.113(0.989–1.251)	0.075	1.085(0.962–1.223)	0.186
Lower lobe	1.105(1.041–1.173)	0.001	1.083(1.019–1.151)	0.010
NOS	1.696(1.586–1.813)	<0.001	1.160(1.068–1.260)	<0.001
Overlapping lesion	1.717(0.942–1.455)	0.156	1.157(0.929–1.440)	0.193
Main bronchus	1.464(1.314–1.631)	<0.001	1.184(1.061–1.322)	0.003
Trachea	1.161(0.521–2.587)	0.714	1.779(0.749–4.277)	0.192
Grade		<0.001		0.049
Grade I	Reference		Reference	
Grade II	0.898(0.492–1.640)	0.726	1.011(0.549–1.861)	0.972
Grade III	1.122(0.650–1.936)	0.679	1.282(0.737–2.229)	0.379
Grade IV	1.190(0.689–2.054)	0.533	1.362(0.783–2.370)	0.274
Unknown	1.548(0.898–2.669)	0.166	1.272(0.732–2.211)	0.394
Stage		<0.001		<0.001
Stage I	Reference		Reference	
Stage II	1.386(1.204–1.597)	<0.001	1.633(1.415–1.884)	<0.001
Stage III	2.497(2.257–2.764)	<0.001	2.088(1.791–2.251)	<0.001
Stage IV	4.764(4.321–5.252)	<0.001	3.115(2.775–3.496)	<0.001
Unknown	3.169(2.780–3.612)	<0.001	1.555(1.342–1.803)	<0.001
Laterality		<0.001		0.004
Right-origin of primary	Reference		Reference	
Left-origin of primary	1.018(0.970–1.070)	0.468	1.033(0.982–1.087)	0.206
Bilateral, single primary	1.906(1.593–2.280)	<0.001	0.949(0.783–1.150)	0.592
Paired, but no laterality	1.434(1.261–1.630)	<0.001	0.781(0.675–0.904)	0.001
Others	1.483(1.131–1.945)	0.004	0.798(0.593–1.075)	0.138
Lymphadenectomy		<0.001		<0.001
0–3 LNRs	Reference		Reference	
≥4 LNRs	0.579(0.511–0.655)	<0.001	0.790(0.692–0.903)	0.001
Biopsy or aspiration	1.597(1.384–1.844)	<0.001	1.079(0.926–1.257)	0.333
Sentinel biopsy	0.556(0.230–1.343)	0.192	0.779(0.321–1.890)	0.581
None	2.013(1.811–2.239)	<0.001	1.162(1.031–1.309)	0.014
Unknown	1.140(0.957–1.359)	0.142	1.077(0.902–1.287)	0.411
Surgery record		<0.001		<0.001
Yes	Reference		Reference	
No	3.107(2.912–3.316)	<0.001	1.714(1.535–1.914)	<0.001
Unknown	3.142(2.346–4.208)	<0.001	1.798(1.323–2.444)	<0.001
Radiation sequence		<0.001		0.009
No radiation and/or surgery	Reference		Reference	
Radiation after surgery	0.747(0.695–0.802)	<0.001	0.913(0.840–0.992)	0.032
Radiation prior to surgery	0.542(0.442–0.665)	<0.001	1.128(0.911–1.396)	0.269
Radiation before and after surgery	1.232(0.512–2.960)	0.641	1.778(0.736–4.298)	0.201
Intraoperative radiation	0.802(0.259–2.487)	0.702	1.040(0.334–3.236)	0.946
Sequence unknown, but both given	1.811(0.863–3.801)	0.116	2.711(1.284–5.724)	0.009
Radiation record		0.766		
Beam radiation	Reference			
Beam with implants or isotopes	0.955(0.477–1.911)	0.896		
Implant or radioisotopes	0.905(0.501–1.635)	0.740		
Radiation, but not specified	1.127(0.840–1.512)	0.426		
No radiation	0.987(0.941–1.035)	0.593		
Unknown	1.135(0.914–1.410)	0.251		
Chemotherapy record		<0.001		<0.001
No/unknown	Reference		Reference	
Yes	0.639(0.610–0.670)	<0.001	0.501(0.477–0.528)	<0.001
Tumor Size		<0.001		<0.001
≤1 cm	Reference		Reference	
>1, ≤2 cm	1.530(0.493–4.747)	0.461	1.678(0.539–5.230)	0.372
>2, ≤3 cm	0.935(0.389–2.249)	0.881	0.595(0.247–1.437)	0.249
>3, ≤4 cm	1.378(0.689–2.757)	0.365	1.334(0.663–2.682)	0.419
>4 cm	1.306(0.490–3.483)	0.593	0.637(0.238–1.703)	0.369
Unknown	1.690(1.602–1.783)	<0.001	1.195(1.122–1.273)	<0.001
Bone Metastasis		<0.001		0.001
No	Reference		Reference	
Yes	2.193(1.981–2.426)	<0.001	1.227(1.101–1.368)	<0.001
Unknown	1.184(1.118–1.254)	<0.001	1.305(0.908–1.877)	0.151
Brain Metastasis		<0.001		0.097
No	Reference		Reference	
Yes	1.777(1.601–1.974)	<0.001	1.110(0.995–1.239)	0.062
Unknown	1.139(1.076–1.206)	<0.001	0.872(0.631–1.205)	0.407
Liver Metastasis		<0.001		<0.001
No	Reference		Reference	
Yes	2.375(2.124–2.656)	<0.001	1.442(1.279–1.624)	<0.001
Unknown	1.149(1.086–1.215)	<0.001	1.082(0.737–1.588)	0.688
Lung Metastasis		<0.001		0.220
No	Reference		Reference	
Yes	1.880(1.673–2.113)	<0.001	1.095(0.969–1.237)	0.147
Unknown	1.117(1.056–1.181)	<0.001	0.887(0.657–1.198)	0.435
First malignant primary indicator		0.001		0.591
No	Reference		Reference	
Yes	1.132(1.053–1.217)	0.001	1.020(0.948–1.098)	0.591
Age at diagnosis		<0.001		<0.001
<65	Reference		Reference	
≥65	1.156(1.103–1.211)	<0.001	1.230(1.171–1.291)	<0.001
Insurance status		<0.001		0.003
Any Medicaid	Reference		Reference	
Insured or no specifics	0.817(0.752–0.888)	<0.001	0.881(0.808–0.961)	0.004
Uninsured	1.063(0.910–1.242)	0.441	1.083(0.925–1.268)	0.321
Unknown	0.875(0.805–0.952)	0.002	0.914(0.833–1.002)	0.056
Marital status		<0.001		<0.001
Married or domestic partner	Reference		Reference	
Divorced/separated/single/widowed	1.187(1.132–1.245)	<0.001	1.121(1.065–1.179)	<0.001
Unknown	1.113(0.974–1.272)	0.114	1.087(0.949–1.245)	0.230
High School education (%)		<0.001		0.681
≤10	Reference		Reference	
>10, ≤20	1.065(0.998–1.137)	0.058	1.017(0.946–1.094)	0.647
>20, ≤30	1.200(1.116–1.290)	<0.001	1.063(0.970–1.165)	0.190
>30	1.195(1.035–1.381)	0.015	1.065(0.900–1.260)	0.463
Unknown	0.955(0.134–6.786)	0.963	1.021(0.143–7.305)	0.984
Median family income (dollar, in tens)		<0.001		0.196
≤5000	Reference		Reference	
>5000, ≤7000	1.019(0.954–1.089)	0.575	1.065(0.987–1.148)	0.104
>7000, ≤9000	0.877(0.814–0.946)	0.001	1.060(0.964–1.165)	0.232
>9000	0.812(0.735–0.898)	<0.001	0.990(0.877–1.118)	0.872
Unknown	0.840(0.118–5.972)	0.862		

Abbreviations: LCC: large cell carcinoma; OS: overall survival; LNRs: lymph nodes removed; NOS: not otherwise specified; HR: hazard ratio; CI: confidence interval. A *p*-value of less than 0.05 represents a significant statistical difference.

**Table 4 cancers-14-05231-t004:** Univariate and multivariate COX regression analyses on OS in LCC patients <65 years old.

Variables	Univariate Analysis		Multivariate Analysis	
	HR (95% CI)	*p*	HR (95% CI)	*p*
Race		0.146		
White	Reference			
Black	1.098(1.007–1.197)	0.034		
Asian and others	0.964(0.812–1.144)	0.672		
Unknown	0.501(0.071–3.599)	0.490		
Sex		<0.001		0.003
Male	Reference		Reference	
Female	0.824(0.769–0.882)	<0.001	0.898(0.837–0.963)	0.003
Year of diagnosis		0.586		
2004–2007	Reference			
2008–2011	0.983(0.911–1.062)	0.667		
2012–2015	1.037(0.944–1.139)	0.448		
Region		0.976		
East	Reference			
Northern Plains	0.995(0.891–1.112)	0.933		
Southwest	0.970(0.791–1.189)	0.767		
Alaska and Pacific Coast	1.011(0.937–1.090)	0.782		
Tumor location		<0.001		0.012
Upper lobe	Reference		Reference	
Middle lobe	1.200(1.013–1.421)	0.035	1.210(1.018–1.440)	0.031
Lower lobe	1.196(1.092–1.309)	<0.001	1.117(1.019–1.225)	0.018
NOS	1.754(1.596–1.929)	<0.001	1.129(0.999–1.276)	0.051
Overlapping lesion	0.968(0.705–1.330)	0.842	0.908(0.659–1.251)	0.554
Main bronchus	1.540(1.332–1.781)	<0.001	1.232(1.062–1.429)	0.006
Trachea	1.740(0.435–6.965)	0.434	1.574(0.358–6.911)	0.548
Grade		<0.001		0.411
Grade I	Reference		Reference	
Grade II	0.733(0.365–1.473)	0.384	0.777(0.381–1.586)	0.489
Grade III	0.858(0.473–1.554)	0.612	1.024(0.556–1.885)	0.941
Grade IV	0.940(0.518–1.703)	0.837	1.083(0.588–1.994)	0.799
Unknown	1.218(0.673–2.204)	0.515	1.031(0.560–1.896)	0.922
Stage		<0.001		<0.001
Stage I	Reference		Reference	
Stage II	1.467(1.187–1.812)	<0.001	1.803(1.455–2.234)	<0.001
Stage III	2.857(2.446–3.336)	<0.001	2.288(1.915–2.734)	<0.001
Stage IV	5.556(4.786–6.450)	<0.001	3.630(3.032–4.346)	<0.001
Unknown	3.584(2.892–4.442)	<0.001	1.836(1.443–2.337)	<0.001
Laterality		<0.001		0.055
Right-origin of primary	Reference		Reference	
Left—origin of primary	1.046(0.974–1.122)	0.218	1.094(1.017–1.177)	0.016
Bilateral, single primary	1.963(1.524–2.530)	<0.001	1.117(0.849–1.471)	0.428
Paired, but no laterality	1.493(1.256–1.776)	<0.001	0.875(0.716–1.070)	0.193
Others	1.747(1.186–2.574)	0.005	0.997(0.657–1.511)	0.988
Lymphadenectomy		<0.001		<0.001
0–3 LNRs	Reference		Reference	
≥4 LNRs	0.523(0.437–0.625)	<0.001	0.707(0.584–0.855)	<0.001
Biopsy or aspiration	1.558(1.274–1.905)	<0.001	0.964(0.778–1.194)	0.738
Sentinel biopsy	0.429(0.137–1.343)	0.146	0.625(0.198–1.971)	0.423
None	1.918(1.647–2.233)	<0.001	1.057(0.890–1.255)	0.527
Unknown	0.987(0.766–1.272)	0.919	0.945(0.729–1.225)	0.670
Surgery record		<0.001		<0.001
Yes	Reference		Reference	
No	3.267(2.974–3.588)	<0.001	1.544(1.317–1.811)	<0.001
Unknown	3.232(2.209–4.729)	<0.001	1.907(1.274–2.854)	0.002
Radiation sequence		<0.001		0.580
No radiation and/or surgery	Reference		Reference	
Radiation after surgery	0.791(0.716–0.873)	<0.001	0.944(0.840–1.062)	0.339
Radiation prior to surgery	0.481(0.358–0.646)	<0.001	0.929(0.680–1.270)	0.646
Radiation before and after surgery	1.213(0.391–3.765)	0.738	1.738(0.557–5.425)	0.341
Intraoperative radiation	0.765(0.108–5.436)	0.789	0.584(0.082–4.174)	0.592
Sequence unknown, but both given	0.956(0.135–6.793)	0.964	3.344(0.466–24.006)	0.230
Radiation record		0.462		
Beam radiation	Reference			
Beam with implants or isotopes	0.854(0.383–1.903)	0.699		
Implant or radioisotopes	1.455(0.652–3.243)	0.360		
Radiation, but not specified	1.008(0.668–1.521)	0.970		
No radiation	0.947(0.885–1.014)	0.120		
Unknown	1.131(0.856–1.495)	0.385		
Chemotherapy record		<0.001		<0.001
No/unknown	Reference		Reference	
Yes	0.639(0.597–0.684)	<0.001	0.524(0.487–0.563)	<0.001
Tumor Size		<0.001		0.011
≤1 cm	Reference		Reference	
>1, ≤2 cm	1.122(0.280–4.490)	0.871	1.183(0.292–4.783)	0.814
>2, ≤3 cm	6.627(0.932–47.124)	0.059	2.749(0.383–19.751)	0.315
>3, ≤4 cm	1.215(0.505–2.921)	0.664	1.321(0.546–3.197)	0.537
Unknown	1.705(1.579–1.841)	<0.001	1.172(1.071–1.284)	0.001
Bone Metastasis		<0.001		0.060
No	Reference		Reference	
Yes	2.247(1.955–2.584)	<0.001	1.189(1.023–1.382)	0.024
Unknown	1.152(1.060–1.251)	0.001	1.358(0.795–2.322)	0.263
Brain Metastasis		<0.001		0.759
No	Reference		Reference	
Yes	1.716(1.489–1.977)	<0.001	1.039(0.894–1.209)	0.616
Unknown	1.095(1.009–1.189)	0.030	0.883(0.522–1.495)	0.644
Liver Metastasis		<0.001		<0.001
No	Reference		Reference	
Yes	2.579(2.217–3.001)	<0.001	1.582(1.342–1.866)	<0.001
Unknown	1.120(1.034–1.214)	0.006	0.919(0.520–1.623)	0.771
Lung Metastasis		<0.001		0.579
No	Reference		Reference	
Yes	2.041(1.743–2.389)	<0.001	1.091(0.923–1.290)	0.309
Unknown	1.085(1.002–1.175)	0.044	0.965(0.589–1.581)	0.889
First malignant primary indicator		<0.001		0.085
No	Reference		Reference	
Yes	1.214(1.091–1.351)	<0.001	1.100(0.987–1.227)	0.085
Insurance status		<0.001		0.002
Any Medicaid	Reference		Reference	
Insured or no specifics	0.746(0.671–0.829)	<0.001	0.851(0.762–0.950)	0.004
Uninsured	1.079(0.913–1.276)	0.372	1.072(0.904–1.272)	0.421
Unknown	0.821(0.739–0.912)	<0.001	0.930(0.826–1.047)	0.231
Marital status		<0.001		0.001
Married or domestic partner	Reference		Reference	
Divorced/separated/single/widowed	1.219(1.139–1.305)	<0.001	1.143(1.064–1.229)	<0.001
Unknown	1.132(0.924–1.386)	0.230	1.040(0.844–1.282)	0.711
High School education (%)		<0.001		0.609
≤10	Reference		Reference	
>10, ≤20	1.085(0.988–1.192)	0.089	1.014(0.914–1.126)	0.790
>20, ≤30	1.270(1.144–1.410)	<0.001	1.079(0.947–1.230)	0.251
>30	1.301(1.060–1.597)	0.012	1.154(0.908–1.466)	0.241
Unknown	1.024(1.144–7.286)	0.981	1.036(0.143–7.487)	0.972
Median family income (dollar, in tens)		<0.001		0.520
≤5000	Reference		Reference	
>5000, ≤7000	1.014(0.922–1.115)	0.773	1.051(0.944–1.171)	0.364
>7000, ≤9000	0.843(0.757–0.939)	0.002	1.015(0.888–1.159)	0.831
>9000	0.748(0.647–0.865)	<0.001	0.959(0.806–1.141)	0.636
Unknown	0.861(0.121–6.124)	0.881		

Abbreviations: LCC: large cell carcinoma; OS: overall survival; LNRs: lymph nodes removed; NOS: not otherwise specified; HR: hazard ratio; CI: confidence interval. A *p*-value of less than 0.05 represents a significant statistical difference.

**Table 5 cancers-14-05231-t005:** Univariate and multivariate COX regression analyses on OS in LCC patients ≥65 years old.

Variables	Univariate Analysis		Multivariate Analysis	
	HR (95% CI)	*p*	HR (95% CI)	*p*
Race		0.044		0.003
White	Reference		Reference	
Black	1.079(0.978–1.191)	0.128	0.948(0.856–1.050)	0.307
Asian and others	0.850(0.738–0.980)	0.025	0.765(0.661–0.885)	<0.001
Unknown	1.173(0.440–3.128)	0.750	1.121(0.419–3.002)	0.820
Sex		0.002		<0.001
Male	Reference		Reference	
Female	0.899(0.842–0.961)	0.002	0.800(0.745–0.859)	<0.001
Year of diagnosis		0.535		
2004–2007	Reference			
2008–2011	0.977(0.907–1.052)	0.540		
2012–2015	0.950(0.866–1.043)	0.280		
Region		0.180		
East	Reference			
Northern Plains	1.082(0.973–1.204)	0.147		
Southwest	1.049(0.861–1.278)	0.633		
Alaska and Pacific Coast	0.957(0.889–1.030)	0.238		
Tumor location		<0.001		0.003
Upper lobe	Reference		Reference	
Middle lobe	1.018(0.865–1.199)	0.826	0.970(0.820–1.148)	0.723
Lower lobe	1.000(0.923–1.084)	0.994	1.056(0.973–1.146)	0.189
NOS	1.618(1.472–1.778)	<0.001	1.189(1.061–1.332)	0.003
Overlapping lesion	1.533(1.136–2.067)	0.005	1.585(1.173–2.142)	0.003
Main bronchus	1.396(1.186–1.643)	<0.001	1.104(0.935–1.303)	0.243
Trachea	0.908(0.340–2.423)	0.848	2.204(0.744–6.529)	0.154
Grade		<0.001		0.136
Grade I	Reference		Reference	
Grade II	1.884(0.449–7.908)	0.387	1.719(0.406–7.278)	0.462
Grade III	2.520(0.628–10.104)	0.192	2.092(0.517–8.464)	0.301
Grade IV	2.572(0.641–10.314)	0.183	2.256(0.557–9.135)	0.254
Unknown	3.376(0.842–13.532)	0.186	2.053(0.508–8.304)	0.313
Stage		<0.001		<0.001
Stage I	Reference		Reference	
Stage II	1.367(1.129–1.655)	0.001	1.503(1.239–1.824)	<0.001
Stage III	2.243(1.961–2.565)	<0.001	1.863(1.604–2.164)	<0.001
Stage IV	4.262(3.743–4.852)	<0.001	2.866(2.462–3.336)	<0.001
Unknown	2.762(2.339–3.261)	<0.001	1.394(1.155–1.682)	0.001
Laterality		<0.001		0.001
Right-origin of primary	Reference		Reference	
Left-origin of primary	0.981(0.917–1.050)	0.584	0.970(0.904–1.040)	0.393
Bilateral, single primary	1.840(1.428–2.369)	<0.001	0.810(0.618–1.061)	0.126
Paired, but no laterality	1.376(1.136–1.668)	0.001	0.669(0.539–0.830)	<0.001
Others	1.237(0.846–1.808)	0.272	0.569(0.369–0.877)	0.011
Lymphadenectomy		<0.001		0.001
0–3 LNRs	Reference		Reference	
≥4 LNRs	0.655(0.552–0.779)	<0.001	0.890(0.738–1.073)	0.222
Biopsy or aspiration	1.686(1.373–2.072)	<0.001	1.201(0.962–1.499)	0.106
Sentinel biopsy	1.036(0.257–4.173)	0.960	0.965(0.238–3.921)	0.961
None	2.162(1.864–2.507)	<0.001	1.267(1.071–1.498)	0.006
Unknown	1.366(1.070–1.742)	0.012	1.231(0.962–1.575)	0.099
Surgery record		<0.001		<0.001
Yes	Reference		Reference	
No	2.977(2.720–3.258)	<0.001	1.900(1.628–2.217)	<0.001
Unknown	3.244(2.052–5.128)	<0.001	1.612(0.998–2.604)	0.051
Radiation sequence		<0.001		0.001
No radiation and/or surgery	Reference		Reference	
Radiation after surgery	0.706(0.636–0.784)	<0.001	0.863(0.765–0.973)	0.016
Radiation prior to surgery	0.635(0.479–0.842)	0.002	1.425(1.059–1.916)	0.019
Radiation before and after surgery	1.306(0.326–5.225)	0.706	1.759(0.431–7.183)	0.431
Intraoperative radiation	0.803(0.201–3.213)	0.757	1.498(0.370–6.068)	0.571
Sequence unknown, but both given	2.039(0.915–4.545)	0.081	2.892(1.283–6.520)	0.010
Radiation record		0.560		
Beam radiation	Reference			
Beam with implants or isotopes	1.677(0.419–6.712)	0.465		
Implant or radioisotopes	0.612(0.254–1.472)	0.273		
Radiation, but not specified	1.280(0.840–1.950)	0.250		
No radiation	1.017(0.952–1.087)	0.621		
Unknown	1.166(0.826–1.646)	0.383		
Chemotherapy record		<0.001		<0.001
No/unknown	Reference		Reference	
Yes	0.645(0.604–0.689)	<0.001	0.474(0.441–0.509)	<0.001
Tumor Size		<0.001		<0.001
≤1 cm	Reference		Reference	
>1, ≤2 cm	11.188(1.572–79.621)	0.016	4.162(0.583–29.731)	0.155
>2, ≤3 cm	0.719(0.269–1.916)	0.509	0.480(0.179–1.287)	0.145
>3, ≤4 cm	2.107(0.679–6.540)	0.197	1.147(0.357–3.690)	0.818
>4 cm	1.236(0.464–3.297)	0.672	0.631(0.236–1.691)	0.360
Unknown	1.674(1.554–1.804)	<0.001	1.224(1.120–1.337)	<0.001
Bone Metastasis		<0.001		0.015
No	Reference		Reference	
Yes	2.160(1.863–2.503)	<0.001	1.269(1.081–1.489)	0.004
Unknown	1.217(1.123–1.319)	<0.001	1.199(0.729–1.973)	0.475
Brain Metastasis		<0.001		0.070
No	Reference		Reference	
Yes	1.903(1.627–2.225)	<0.001	1.193(1.013–1.406)	0.035
Unknown	1.186(1.096–1.284)	<0.001	0.874(0.576–1.324)	0.524
Liver Metastasis		<0.001		0.010
No	Reference		Reference	
Yes	2.186(1.852–2.581)	<0.001	1.315(1.101–1.572)	0.003
Unknown	1.178(1.089–1.274)	<0.001	1.225(0.718–2.090)	0.457
Lung Metastasis		<0.001		0.506
No	Reference		Reference	
Yes	1.721(1.446–2.049)	<0.001	1.076(0.895–1.293)	0.436
Unknown	1.149(1.063–1.242)	<0.001	0.865(0.589–1.270)	0.459
First malignant primary indicator		0.239		
No	Reference			
Yes	1.061(0.961–1.171)	0.239		
Insurance status		0.063		
Any Medicaid	Reference			
Insured or no specifics	0.860(0.746–0.992)	0.038		
Uninsured	1.046(0.598–1.832)	0.874		
Unknown	0.924(0.799–1.067)	0.281		
Marital status		<0.001		0.007
Married or domestic partner	Reference		Reference	
Divorced/separated/single/widowed	1.154(1.079–1.234)	<0.001	1.121(1.042–1.205)	0.002
Unknown	1.087(0.911–1.298)	0.355	1.127(0.940–1.351)	0.198
High School education (%)		0.089		
≤10	Reference			
>10, ≤20	1.044(0.953–1.143)	0.356		
>20, ≤30	1.128(1.021–1.248)	0.018		
>30	1.089(0.888–1.335)	0.413		
Unknown				
Median family income (dollar, in tens)		0.012		0.360
≤5000	Reference		Reference	
>5000, ≤7000	1.026(0.934–1.126)	0.591	1.077(0.979–1.184)	0.127
>7000, ≤9000	0.917(0.826–1.019)	0.109	1.072(0.961–1.195)	0.211
>9000	0.886(0.771–1.018)	0.087	1.006(0.874–1.159)	0.932

Abbreviations: LCC: large cell carcinoma; OS: overall survival; LNRs: lymph nodes removed; NOS: not otherwise specified; HR: hazard ratio; CI: confidence interval. A *p*-value of less than 0.05 represents a significant statistical difference.

## Data Availability

All data generated during this study are included in this article. The datasets supporting the conclusions of this article are available in SEER database: https://seer.cancer.gov/, accessed on 10 April 2022.

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
