# Peer review of "The Clinical Characteristics and Treatments for Large Cell Carcinoma Patients Older than 65 Years Old: A Population-Based Study"

_cancers, 2022, doi:10.3390/cancers14215231_

Round 1
Reviewer 1 Report
This is an original manuscript about treating large cell lung cancer with radiotherapy as sequential therapy with surgery. Although it is a retrospective article, a large number of patients were involved in their analysis, and they were divided into two groups, above 65 years old and below. The Authors did a great job, so I congratulate them on their work. But I have some concerns about the radiotherapy part.
It will be exciting for the reader to know which Dose concept was used to treat the patients both for postoperative as also neoadjuvant radiotherapy. Since the neoadjuvant concept does not affect the OS, the question was what was the treatment concept applied to these patients.
Since the authors mentioned the different technologies of radiotherapy, so the dose concept will also belong to the description in the methods and materials and the discussion. And also, since the time of the patient's collection was almost ten years, and there was considerable development in the field of radiotherapy, external beam radiotherapy was used 3D-Planning or IMRT. The authors should also explain which isotope was used in Implantation (Iodine, palladium)
Author Response
Response to Reviewer 1Comments
Dear editors and reviewers,
Thank you for your careful review and valuable suggestions of our manuscript entitled “Radiotherapy sequences with surgery for stage III/IV large cell carcinoma patients older than 65 years old: A population-based study”. The following are our point-by-point responses to reviewers and editors’ comments. The comments are shown in italics and are followed by our responses.
Editors:
Thank you very much for your work. I have modified the paper according to your requirements.
Reviewer 1:
This is an original manuscript about treating large cell lung cancer with radiotherapy as sequential therapy with surgery. Although it is a retrospective article, a large number of patients were involved in their analysis, and they were divided into two groups, above 65 years old and below. The Authors did a great job, so I congratulate them on their work. But I have some concerns about the radiotherapy part.
Point 1: It will be exciting for the reader to know which Dose concept was used to treat the patients both for postoperative as also neoadjuvant radiotherapy. Since the neoadjuvant concept does not affect the OS, the question was what was the treatment concept applied to these patients.
Point 2: Since the authors mentioned the different technologies of radiotherapy, so the dose concept will also belong to the description in the methods and materials and the discussion. And also, since the time of the patient's collection was almost ten years, and there was considerable development in the field of radiotherapy, external beam radiotherapy was used 3D-Planning or IMRT. The authors should also explain which isotope was used in Implantation (Iodine, palladium)
Response 1 & 2: Thank you very much for your efforts on our work. The advice you raised is very meaningful. It was very regrettable that the specific dose of radiotherapy and the specific types of radioisotopes were not explicitly represented in the SEER database. Besides postoperative radiotherapy, we also found lymphadenectomy, surgery and chemotherapy were all beneficial to prognosis of patients. Because of the important development of radiotherapy in the recent years, it is necessary to add new radiation methods in the discussion (please see page 8, line 266 in the modified version). We will carry out this project in the future subsequent clinical trials.
Please see the attachment.

Reviewer 2 Report
In this study, the authors dissected in depth the impact of clinical characteristics and the therapeutic strategies on the survival outcomes in LCC patients. The paper is well written. No changes are requested.
Author Response
Response to Reviewer 2Comments
Dear editors and reviewers,
Thank you for your careful review and valuable suggestions of our manuscript entitled “Radiotherapy sequences with surgery for stage III/IV large cell carcinoma patients older than 65 years old: A population-based study”. The following are our point-by-point responses to reviewers and editors’ comments. The comments are shown in italics and are followed by our responses.
Editors:
Thank you very much for your work. I have modified the paper according to your requirements.
Reviewer 2:
In this study, the authors dissected in depth the impact of clinical characteristics and the therapeutic strategies on the survival outcomes in LCC patients. The paper is well written. No changes are requested.
Response: Thank you very much for your affirmation. Some adjustments have been made to meet the requirement of word count, which can be seen in the modified version. We have reviewed all the requirements and confirm that our revision meet the requirements of the magazine.
Please see the attachment
Reviewer 3 Report
Thank you for the opportunity to review the study. Below please see my detailed comments and suggestions.
1. According to the authors, there is a lack of information on LCC about patient and clinical factors and appropriate treatment. Therefore, they conducted this retrospective study to explore how different factors and treatment affect survival of stage III/IV LCC patients. If this is the aim of this study, then there is no need to match to other NSCLC patients and compare the matched samples. Doing so will restrict the LCC patients to a group similar to other NSCLC patients, which would limit the generalizability of the study. The authors should just focus on the LCC patients only.
2. The authors stated that the aim is to study this in stage III/IV LCC patients. However, all patients with stages 1-4 are included.
3. Data were from 2004-2015, 7-18 years ago. Lung cancer treatment has gone through significant changes in the last decade, especially with the immunotherapy use. The FDA-approval for these agents mostly came on 2015 and after, making the findings in this study, especially the treatment information, less relevant.
3. Propensity score matching method was not described sufficiently. What variables were used in matching? How were groups matched? What software was used for the matching? How was balancing in baseline characteristics determined? Also, the authors should include a table (at least as a supplement) that shows the balancing of covariates. But, again, for the study aim, PS matching is not needed.
4. SEER now represents 48% of population. Is 25% for SEER 2004-2015? Please check SEER website for the latest number.
5. It’s not clear why the authors excluded patients “absence of organ metastasis” since the study included stage IV patients. Also, how did the authors determine the presence of organ metastasis? Does SEER have metastasis information to other organs other than lung, liver, and brain? Based on my knowledge, brain metastasis information become available only in 2010 and after. Also, in a lung cancer patient population, what does having lung metastasis mean?
6. In SEER, information on chemotherapy may not be complete. https://seer.cancer.gov/data-software/documentation/seerstat/nov2021/treatment-limitations-nov2021.html
7. “No cancer-direct treatment”. How did the authors define this? Please see my comment above.
8. What is the “first malignant primary indicator”? In Figure 1, patients with prior malignance have been excluded. Shouldn’t all patients have a value of 1 for this variable (i.e. lung cancer should be the first primary cancer for all patients in the study)?
9. The categories for school education and income variables are hard to interpret. From what I know, these variables should be characteristics of the census track where the patients reside, not patients’ own education level or family income level. Please clearly define those variables in the method section.
10. In Table 3, it is not clear what are p-values for and how to interpret them.
2. Line 53. Is this American Cancer Center or American Cancer Society?
Author Response
Response to Reviewer 3Comments
Dear editors and reviewers,
Thank you for your careful review and valuable suggestions of our manuscript entitled “Radiotherapy sequences with surgery for stage III/IV large cell carcinoma patients older than 65 years old: A population-based study”. The following are our point-by-point responses to reviewers and editors’ comments. The comments are shown in italics and are followed by our responses.
Editors:
Thank you very much for your work. I have modified the paper according to your requirements.
Reviewer 3: Thank you for the opportunity to review the study. Below please see my detailed comments and suggestions.
Point1: According to the authors, there is a lack of information on LCC about patient and clinical factors and appropriate treatment. Therefore, they conducted this retrospective study to explore how different factors and treatment affect survival of stage III/IV LCC patients. If this is the aim of this study, then there is no need to match to other NSCLC patients and compare the matched samples. Doing so will restrict the LCC patients to a group similar to other NSCLC patients, which would limit the generalizability of the study. The authors should just focus on the LCC patients only.
Response1: Thank you for your constructive suggestions. Your suggestion is very to the point. Because the LCC patients were screened from the NSCLC patients in SEER database, we would like to know the differences between the LCC patients and other types of NSCLC patients. Because LCC was a type of rare NSCLC which lacked clinical information of patients, it was necessary to compare the characteristics and prognosis of the two groups. We found LCC patients had significantly worse survival outcomes than other types of NSCLC patients, which may be helpful to the future researches of LCC patients.
Point 2: The authors stated that the aim is to study this in stage III/IV LCC patients. However, all patients with stages 1-4 are included.
Response 2: Thank you very much for your efforts on our work. The most of LCC patients were stage III/IV patients (76.7%), so we would like to know if there was correlation between the advanced tumor stage and poor prognosis of LCC patients. In addition, LCC patients had apparently more advanced stage patients than other types of NSCLC patients (30.9%), and we found LCC patients had significantly worse survival time than other types of NSCLC patients. We will carry out this project in the future subsequent clinical trials.
Point 3: Data were from 2004-2015, 7-18 years ago. Lung cancer treatment has gone through significant changes in the last decade, especially with the immunotherapy use. The FDA-approval for these agents mostly came on 2015 and after, making the findings in this study, especially the treatment information, less relevant.
Response 3: Thank you very much for your meaningful advice. This is a very constructive suggestion to improve the quality of the research. Our study mainly focuses on the efficacy of surgical radiotherapy sequence for stage III / IV large cell lung cancer, so the exploration of immunotherapy is not the focus of our study. And it was very regrettable that the immunotherapy was not represented in the SEER database. But due to the considerable developments of the immunotherapy of LCC patients in the recent years, we had relevant discussion in our article (please see page 7, line 252 in the modified version). It was very necessary to explore the immunotherapy of LCC patients in the future subsequent clinical trials.
Point 4: Propensity score matching method was not described sufficiently. What variables were used in matching? How were groups matched? What software was used for the matching? How was balancing in baseline characteristics determined? Also, the authors should include a table (at least as a supplement) that shows the balancing of covariates. But, again, for the study aim, PS matching is not needed.
Response 4: Thank you very much for your constructive suggestions. Your suggestion is very to the point. Due to the apparently quantitative distinctions between different groups, we chose the propensity matching score analysis to make the two cohorts less biased. We had PSM twice between different groups successively by IBM SPSS, version 25.0, in which 1:1 without replacement and nearest-neighbor matching method of 0.02 was used. Firstly, we had PSM between the LCC group (n=11349) and other types of NSCLC group (n=129118) which generated 11349 pairs after PSM, then we performed Kaplan–Meier analysis between the two groups. Thank you for your reminding that we should show the balancing of covariates, so we added relevant analysis after PSM in Table S1 (please see Table S1 in supplement). Secondly, we had PSM between the LCC group <65 years old (n=4300) and LCC group ≥65 years old (n=7049) which generated 4068 pairs after PSM, then we performed Kaplan–Meier analysis and COX regression analysis between the two groups.
Table S1. Baseline characteristics of patients with LCC and other types of NSCLC after PSM
|
Characteristics
|
LCC (n=11349) |
Others (n=11349) |
|
Race White Black Asian and others Unknown |
9243(81.4%) 1568(13.8%) 528(4.7%) 10(0.1%) |
9312(82.1%) 1437(12.7%) 582(5.1%) 18(0.2%) |
|
Sex Male Female |
6618(58.3%) 4731(41.7%) |
6720(59.2%) 4629(40.8%) |
|
Year of diagnosis 2004-2007 2008-2011 2012-2015 |
5487(48.3%) 3508(30.9%) 2354(20.7%) |
4624(40.7%) 3764(33.2%) 2961(26.1%) |
|
Region East Northern Plains Southwest Alaska and Pacific Coast |
5979(52.7%) 1237(10.9%) 343(3.0%) 3790(33.4%) |
5729(50.5%) 1346(11.9%) 313(2.8%) 3961(34.9%) |
|
Tumor location Upper lobe Middle lobe Lower lobe NOS Overlapping lesion Main bronchus Trachea |
5857(51.6%) 465(4.1%) 2573(22.7%) 1761(15.5%) 137(1.2%) 546(4.8%) 10(0.1%) |
5729(50.5%) 413(3.6%) 3147(27.7%) 1260(11.1%) 144(1.3%) 612(5.4%) 23(0.2%) |
|
Grade Grade I Grade II Grade III Grade IV Unknown |
28(0.2%) 95(0.8%) 3368(29.7%) 2781(24.5%) 5077(44.7%) |
364(3.2%) 1862(16.4%) 2731(24.1%) 106(0.9%) 6286(55.4%) |
|
Stage Stage I Stage II Stage III Stage IV Unknown |
1269(11.2%) 668(5.9%) 2972(26.2%) 5735(50.5%) 705(6.2%) |
1501(13.2%) 810(7.1%) 3572(31.5%) 4423(39.0%) 1043(9.2%) |
|
Laterality Right-origin of primary Left - origin of primary Bilateral, single primary Paired, but no laterality Others |
6282(55.4%) 4420(38.9%) 172(1.5%) 401(3.5%) 74(0.7%) |
6162(54.3%) 4698(41.4%) 142(1.3%) 268(2.4%) 79(0.7%) |
|
Lymphadenectomy 0-3 LNRs ≥4 LNRs Biopsy or aspiration Sentinel biopsy None Unknown |
585(5.2%) 1683(14.8%) 542(4.8%) 14(0.1%) 8216(72.4%) 309(2.7%) |
531(4.7%) 1883(16.6%) 542(4.8%) 17(0.1%) 8081(71.2%) 295(2.6%) |
|
Surgery record Yes No Unknown |
2419(21.3%) 8854(78.0%) 76(0.7%) |
2655(23.4%) 8585(75.6%) 109(1.0%) |
|
Radiation sequence No radiation and/or surgery Radiation after surgery Radiation prior to surgery Radiation before and after S Intraoperative radiation Sequence unknown, but both given Surgery before and after radiation Radiation in and before/after surgery |
9951(87.7%) 1218(10.7%) 156(1.4%) 12(0.1%) 3(0.0%) 14(0.1%) 0(0.0%) 0(0.0%) |
10076(88.8%) 1070(9.4%) 163(1.4%) 23(0.2%) 4(0.0%) 8(0.1%) 3(0.0%) 2(0.0%) |
|
Radiation record Beam radiation Beam with implants or isotopes Implant or radioisotopes Radiation, but not specified No radiation Unknown |
4680(41.2%) 8(0.1%) 18(0.2%) 58(0.5%) 6445(56.8%) 140(1.2%) |
4727(41.7%) 19(0.2%) 19(0.2%) 79(0.7%) 6399(56.4%) 106(0.9%) |
|
Chemotherapy record Yes No |
5408(47.7%) 5941(52.3%) |
5293(46.6%) 6056(53.4%) |
|
Tumor Size ≤1cm >1, ≤2cm >2, ≤3cm >3, ≤4cm >4 cm Unknown |
8567(75.5%) 7(0.1%) 9(0.1%) 9(0.1%) 6(0.1%) 2751(24.2%) |
8667(76.4%) 4(0.0%) 12(0.1%) 6(0.1%) 3(0.0%) 2657(23.4%) |
|
Bone Metastasis Yes No Unknown |
759(6.7%) 2937(25.9%) 7653(67.4%) |
613(5.4%) 3829(33.7%) 6907(60.9%) |
|
Brain Metastasis Yes No Unknown |
668(5.9%) 3024(26.6%) 7657(67.5%) |
390(3.4%) 4034(35.5%) 6925(61.0%) |
|
Liver Metastasis Yes No Unknown |
610(5.4%) 3086(27.2%) 7653(67.4%) |
374(3.3%) 4064(35.8%) 6911(60.9%) |
|
Lung Metastasis Yes No Unknown |
551(4.9%) 3125(27.5%) 7673(67.6%) |
557(4.9%) 3896(34.3%) 6896(60.8%) |
|
First malignant primary indicator Yes No |
9213(81.2%) 2136(18.8%) |
8807(77.6%) 2542(22.4%) |
|
Age at diagnosis <65 ≥65 |
4300(37.9%) 7049(62.1%) |
3819(33.7%) 7530(66.3%) |
|
Insurance status Any Medicaid Insured or no specifics Uninsured Unknown |
993(8.7%) 5655(49.8%) 4441(39.1%) |
1052(9.3%) 6368(56.1%) 240(2.1%) 3689(32.5%) |
|
Marital status Married or domestic partner Divorced/separated/single/widowed Unknown |
5975(52.6%) 4995(44.0%) 379(3.3%) |
5922(52.2%) 4993(44.0%) 434(3.8%) |
|
High school education (%) ≤10 >10, ≤20 >20, ≤30 >30 Unknown |
2050(18.1%) 5984(52.7%) 2981(26.3%) 332(2.9%) 2(0.0%) |
2202(19.4%) 5822(51.3%) 2929(25.8%) 395(3.5%) 1(0.0%) |
|
Median family income (dollar, in tens) ≤5000 >5000, ≤7000 >7000, ≤9000 >9000 Unknown |
1682(14.8%) 5745(50.6%) 2854(25.1%) 1066(9.4%) 2(0.0%) |
1682(14.8%) 5470(48.2%) 2909(25.6%) 1287(11.3%) 1(0.0%) |
Point 5: SEER now represents 48% of population. Is 25% for SEER 2004-2015? Please check SEER website for the latest number.
Response 5: Thank you for your reminding. We are very sorry for the mistakes. SEER database represented 28% of population from 2004-2015. We modified the population proportion for SEER database in the modified version (please see page 3, line 92). Surveillance, Epidemiology and End-Results program Web site available at https://seer.cancer.gov/data/seerstat/nov2014/.html
Point 6: It’s not clear why the authors excluded patients “absence of organ metastasis” since the study included stage IV patients. Also, how did the authors determine the presence of organ metastasis? Does SEER have metastasis information to other organs other than lung, liver, and brain? Based on my knowledge, brain metastasis information become available only in 2010 and after. Also, in a lung cancer patient population, what does having lung metastasis mean?
Response 6: Thank you very much for your reminding. We are very sorry for the mistakes in the flow diagram. We deleted the “absence of organ metastasis” to show the correct screening process (please see modified Figure 1). It was very regrettable that more information about organ metastasis were not explicitly represented in the SEER database. Please see website http://seer.cancer.gov/seerstat/variables/seer/ajcc-stage. In addition, Patients enrolled in our study included primary lung cancer patients as well as metastatic lung cancer patients, so “lung metastasis” means metastatic lung cancer patients.
Point 7: In SEER, information on chemotherapy may not be complete. https://seer.cancer.gov/data-software/documentation/seerstat/nov2021/treatment-limitations-nov2021.html
Response 7: Thank you very much for your reminding. We are very sorry for the mistakes. We checked the tables and found that “No” should be replaced by “No/unknown”. But it was a pity that more specific information about chemotherapy was not recorded in detail in SEER database. We will carry out this project in the future subsequent clinical trials.
Point 8: “No cancer-direct treatment”. How did the authors define this? Please see my comment above.
Response 8: Thank you very much for your reminding. We are very sorry for the mistakes in the flow diagram. We replaced it by “without any cancer-directed treatment” to show the correct screening process (please see modified Figure 1, and page 4, line 98 in modified version).
Point 9: What is the “first malignant primary indicator”? In Figure 1, patients with prior malignance have been excluded. Shouldn’t all patients have a value of 1 for this variable (i.e.lung cancer should be the first primary cancer for all patients in the study)?
Response 9: Thank you very much for your reminding. We are very sorry for the mistakes in the flow diagram. We deleted the “malignant tumor history” to show the correct screening process (please see modified Figure 1, and page 4, line 98 in modified version). Patients enrolled in our study included primary lung cancer patients as well as metastatic lung cancer patients.
Point 10: The categories for school education and income variables are hard to interpret. From what I know, these variables should be characteristics of the census track where the patients reside, not patients’ own education level or family income level. Please clearly define those variables in the method section.
Response 10: Thank you very much for your reminding. We are very sorry for the mistakes in tables and modified them already (please see modified Table 1-5 in modified version). Similar to the lecture 1 and lecture 2, the high school education (%) and the median family income (dollars, in tens) in our article represented characteristics of the census track where the patients reside.
1.Hastert, T.A.; Ruterbusch, J.J.; Beresford, S.A.; Sheppard, L.; White, E. Contribution of health behaviors to the association between area-level socioeconomic status and cancer mortality. Social science & medicine (1982) 2016, 148, 52-58, doi:10.1016/j.socscimed.2015.11.023.
2.Varlotto, J.M.; Voland, R.; McKie, K.; Flickinger, J.C.; DeCamp, M.M.; Maddox, D.; Rava, P.; Fitzgerald, T.J.; Graeber, G.; Rassaei, N.; et al. Population-based differences in the outcome and presentation of lung cancer patients based upon racial, histologic, and economic factors in all lung patients and those with metastatic disease. Cancer medicine 2018, 7, 1211-1220, doi:10.1002/cam4.1430.
Point 11: In Table 3, it is not clear what are p-values for and how to interpret them.
Response 11: Thank you very much for your reminding. In Table 3, we performed the univariate COX regression analysis for each variable, and we did the multivariate cox regression survival analysis again. The description that A p-value < 0.05 means that this variable has an effect on overall survival was added below tables (please see modified Table 3-5).
Point 12: Line 53. Is this American Cancer Center or American Cancer Society?
Response 12: Thank you very much for your reminding. I am very sorry for this mistake. I have made corrections. “According to the American Cancer Center” is replaced by “According to the American Cancer Society” (Please see Page 3, Line 54 in modified version).
Please see the attachment

Round 2
Reviewer 1 Report
The Authors considered in their new amnuscript the notes and correction of the the reviwer and I recommended now to publish the new manuscript. This manuscript about the benefit of postoperative radiotherapy in Stage III/IV patients with lung cancer especially large cell carcinoma.
Author Response
Response to Reviewer 1Comments
Dear editors and reviewers,
Thank you for your careful review and valuable suggestions of our manuscript entitled “Radiotherapy sequences with surgery for stage III/IV large cell carcinoma patients older than 65 years old: A population-based study”. The following are our point-by-point responses to reviewers and editors’ comments. The comments are shown in italics and are followed by our responses.
Editors:
Thank you very much for your work. I have modified the paper according to your requirements.
Reviewer 1:
The Authors considered in their new manuscript the notes and correction of the reviewer and I recommended now to publish the new manuscript. This manuscript about the benefit of postoperative radiotherapy in Stage III/IV patients with lung cancer especially large cell carcinoma.
Response: Thank you very much for your affirmation. Some other adjustments have been made to meet the requirements of the magazine, which can be seen in the modified version. We have reviewed all the requirements and confirm that our revision meet the requirements of the magazine.
Please see the attachment.

Reviewer 3 Report
Thank you for responding to my comments. I have several additional concerns.
1. There are still imbalance after matching in key characteristics associated with survival (e.g. grade, stage). How did the authors determined whether the balance was achieved? Usually standardized differences are calculated after matching. I don’t see that in the added table.
2. It is not clear from the tables that education and family income variables are at census track level. Please make this clear in the method section.
3. Please add the information of the test used to generate the bolded p-values from the multivariate analysis in the method section.
4. Although the title indicated that radiation sequence is the focus, the study did not focus on that. For instance, in the discussion section on page 18, “In this study, we mainly explored the impact of clinical characteristics and the therapeutic strategies on the survival outcomes in LCC patients, especially the elderly patients.” This did not mention anything about radiation sequence. I think the title should be changed to reflect what this article is really about.
5. The sample sizes in radiation sequence categories other than before and after surgery are too small to make valid inferences about.
6. The table showing the distributions of the variables after matching should be included in the manuscript as a supplemental table at least.
Author Response
Dear editors and reviewers,
Thank you for your careful review and valuable suggestions of our manuscript entitled “Radiotherapy sequences with surgery for stage III/IV large cell carcinoma patients older than 65 years old: A population-based study”. The following are our point-by-point responses to reviewers and editors’ comments. The comments are shown in italics and are followed by our responses.
Editors:
Thank you very much for your work. I have modified the paper according to your requirements.
Reviewer 3:
Thank you for responding to my comments. I have several additional concerns.
Point 1. There are still imbalances after matching in key characteristics associated with survival (e.g. grade, stage). How did the authors determined whether the balance was achieved? Usually standardized differences are calculated after matching. I don’t see that in the added table.
Response 1: Thank you very much for your efforts on our work. The advice you raised is very meaningful. We added standardized differences before and after matching (please see modified Table 1 & S1 in modified version and supplemental attachments). Thanks again for your suggestions.
Table 1. Baseline characteristics of patients with LCC and other types of NSCLC before PSM.
|
Characteristics
|
LCC (n=11349) |
Others (n=129118) |
P |
|
Race White Black Asian and others Unknown |
9243(81.4%) 1568(13.8%) 528(4.7%) 10(0.1%) |
106372(82.4%) 15398(11.9%) 7170(5.6%) 178(0.1%) |
<0.001 |
|
Sex Male Female |
6618(58.3%) 4731(41.7%) |
78075(60.5%) 51043(39.5%) |
<0.001 |
|
Year of diagnosis 2004-2007 2008-2011 2012-2015 |
5487(48.3%) 3508(30.9%) 2354(20.7%) |
39869(30.9%) 43981(34.1%) 45268(35.1%) |
<0.001 |
|
Region East Northern Plains Southwest Alaska and Pacific Coast |
5979(52.7%) 1237(10.9%) 343(3.0%) 3790(33.4%) |
61337(47.5%) 14743(11.4%) 3663(2.8%) 49375(38.2%) |
<0.001 |
|
Tumor location Upper lobe Middle lobe Lower lobe NOS Overlapping lesion Main bronchus Trachea |
5857(51.6%) 465(4.1%) 2573(22.7%) 1761(15.5%) 137(1.2%) 546(4.8%) 10(0.1%) |
65624(50.8%) 4843(3.8%) 37529(29.1%) 11856(9.2%) 1817(1.4%) 7147(5.5%) 302(0.2%) |
<0.001 |
|
Grade Grade I Grade II Grade III Grade IV Unknown |
28(0.2%) 95(0.8%) 3368(29.7%) 2781(24.5%) 5077(44.7%) |
7401(5.7%) 33817(26.2%) 39050(30.2%) 888(0.7%) 47961(37.1%) |
<0.001 |
|
Stage Stage I Stage II Stage III Stage IV Unknown |
1269(11.2%) 668(5.9%) 2972(26.2%) 5735(50.5%) 705(6.2%) |
23033(17.8%) 10475(8.1%) 40502(31.4%) 45210(35.0%) 9898(7.7%) |
<0.001 |
|
Laterality Right-origin of primary Left - origin of primary Bilateral, single primary Paired, but no laterality Others |
6282(55.4%) 4420(38.9%) 172(1.5%) 401(3.5%) 74(0.7%) |
71226(55.2%) 53577(41.5%) 1401(1.1%) 2184(1.7%) 730(0.6%) |
<0.001 |
|
Lymphadenectomy 0-3 LNRs ≥4 LNRs Biopsy or aspiration Sentinel biopsy None Unknown |
585(5.2%) 1683(14.8%) 542(4.8%) 14(0.1%) 8216(72.4%) 309(2.7%) |
5949(4.6%) 25491(19.7%) 5829(4.5%) 246(0.2%) 88610(68.6%) 2993(2.3%) |
<0.001 |
|
Surgery record Yes No Unknown |
2419(21.3%) 8854(78.0%) 76(0.7%) |
35272(27.3%) 124529(71.9%) 1062(0.8%) |
<0.001 |
|
Radiation sequence No radiation and/or surgery Radiation after surgery Radiation prior to surgery Radiation before and after S Intraoperative radiation Sequence unknown, but both given Surgery before and after radiation Radiation in and before/after surgery |
9951(87.7%) 1218(10.7%) 156(1.4%) 12(0.1%) 3(0.0%) 14(0.1%) 0(0.0%) 0(0.0%) |
115975(89.8%) 11180(8.7%) 1611(1.2%) 197(0.2%) 27(0.0%) 88(0.1%) 28(0.0%) 12(0.0%) |
<0.001 |
|
Radiation record Beam radiation Beam with implants or isotopes Implant or radioisotopes Radiation, but not specified No radiation Unknown |
4680(41.2%) 8(0.1%) 18(0.2%) 58(0.5%) 6445(56.8%) 140(1.2%) |
52573(40.7%) 204(0.2%) 269(0.2%) 777(0.6%) 74095(57.4%) 1200(0.9%) |
0.002 |
|
Chemotherapy record Yes No/unknown |
5408(47.7%) 5941(52.3%) |
57335(44.4%) 71783(55.6%) |
0.001 |
|
Tumor Size ≤1cm >1, ≤2cm >2, ≤3cm >3, ≤4cm >4 cm Unknown |
8567(75.5%) 7(0.1%) 9(0.1%) 9(0.1%) 6(0.1%) 2751(24.2%) |
102975(79.8%) 57(0.0%) 102(0.1%) 100(0.1%) 62(0.0%) 25822(20.0%) |
<0.001 |
|
Bone Metastasis Yes No Unknown |
759(6.7%) 2937(25.9%) 7653(67.4%) |
8056(6.2%) 57178(44.3%) 63884(49.5%) |
<0.001 |
|
Brain Metastasis Yes No Unknown |
668(5.9%) 3024(26.6%) 7657(67.5%) |
4139(3.2%) 61029(47.3%) 63950(49.5%) |
<0.001 |
|
Liver Metastasis Yes No Unknown |
610(5.4%) 3086(27.2%) 7653(67.4%) |
4099(3.2%) 61089(47.3%) 63930(49.5%) |
<0.001 |
|
Lung Metastasis Yes No Unknown |
551(4.9%) 3125(27.5%) 7673(67.6%) |
8226(6.4%) 56758(44.0%) 64134(49.7%) |
<0.001 |
|
First malignant primary indicator Yes No |
9213(81.2%) 2136(18.8%) |
98320(76.1%) 30798(23.9%) |
<0.001 |
|
Age at diagnosis <65 ≥65 |
4300(37.9%) 7049(62.1%) |
37481(29.0%) 91636(71.0%) |
<0.001 |
|
Insurance status Any Medicaid Insured or no specifics Uninsured Unknown |
993(8.7%) 5655(49.8%) 4441(39.1%) |
13440(10.4%) 81576(63.2%) 2372(1.8%) 31730(24.6%) |
<0.001 |
|
Marital status Married or domestic partner Divorced/separated/single/widowed Unknown |
5975(52.6%) 4995(44.0%) 379(3.3%) |
66147(51.2%) 57548(44.6%) 5423(4.2%) |
<0.001 |
|
High School education (%) ≤10 >10, ≤20 >20, ≤30 >30 Unknown |
2050(18.1%) 5984(52.7%) 2981(26.3%) 332(2.9%) 2(0.0%) |
26234(20.3%) 66441(51.5%) 32302(25.0%) 4128(3.2%) 13(0.0%) |
<0.001 |
|
Median Family income (dollar, in tens) ≤5000 >5000, ≤7000 >7000, ≤9000 >9000 Unknown |
1682(14.8%) 5745(50.6%) 2854(25.1%) 1066(9.4%) 2(0.0%) |
16744(13.0%) 62365(48.3%) 34034(26.4%) 15962(12.4%) 13(0.0%) |
<0.001 |
Abbreviations: NSCLC: non-small cell lung cancer; LCC: large cell carcinoma. The bolded p-value of less than 0.05 represents a significant statistical difference.
Table S1. Baseline characteristics of patients with LCC and other types of NSCLC after PSM
|
Characteristics
|
LCC (n=11349) |
Others (n=11349) |
P |
|
|
|
|
|
|
Race White Black Asian and others Unknown |
9243(81.4%) 1568(13.8%) 528(4.7%) 10(0.1%) |
9312(82.1%) 1437(12.7%) 582(5.1%) 18(0.2%) |
0.012 |
|
|
|
|
|
|
Sex Male Female |
6618(58.3%) 4731(41.7%) |
6720(59.2%) 4629(40.8%) |
0.169 |
|
|
|
|
|
|
Year of diagnosis 2004-2007 2008-2011 2012-2015 |
5487(48.3%) 3508(30.9%) 2354(20.7%) |
4624(40.7%) 3764(33.2%) 2961(26.1%) |
<0.001 |
|
|
|
|
|
|
Region East Northern Plains Southwest Alaska and Pacific Coast |
5979(52.7%) 1237(10.9%) 343(3.0%) 3790(33.4%) |
5729(50.5%) 1346(11.9%) 313(2.8%) 3961(34.9%) |
0.002 |
|
|
|
|
|
|
Tumor location Upper lobe Middle lobe Lower lobe NOS Overlapping lesion Main bronchus Trachea |
5857(51.6%) 465(4.1%) 2573(22.7%) 1761(15.5%) 137(1.2%) 546(4.8%) 10(0.1%) |
5729(50.5%) 413(3.6%) 3147(27.7%) 1260(11.1%) 144(1.3%) 612(5.4%) 23(0.2%) |
<0.001 |
|
|
|
|
|
|
Grade Grade I Grade II Grade III Grade IV Unknown |
28(0.2%) 95(0.8%) 3368(29.7%) 2781(24.5%) 5077(44.7%) |
364(3.2%) 1862(16.4%) 2731(24.1%) 106(0.9%) 6286(55.4%) |
<0.001 |
|
|
|
|
|
|
Stage Stage I Stage II Stage III Stage IV Unknown |
1269(11.2%) 668(5.9%) 2972(26.2%) 5735(50.5%) 705(6.2%) |
1501(13.2%) 810(7.1%) 3572(31.5%) 4423(39.0%) 1043(9.2%) |
<0.001 |
|
|
|
|
|
|
Laterality Right-origin of primary Left - origin of primary Bilateral, single primary Paired, but no laterality Others |
6282(55.4%) 4420(38.9%) 172(1.5%) 401(3.5%) 74(0.7%) |
6162(54.3%) 4698(41.4%) 142(1.3%) 268(2.4%) 79(0.7%) |
<0.001 |
|
|
|
|
|
|
Lymphadenectomy 0-3 LNRs ≥4 LNRs Biopsy or aspiration Sentinel biopsy None Unknown |
585(5.2%) 1683(14.8%) 542(4.8%) 14(0.1%) 8216(72.4%) 309(2.7%) |
531(4.7%) 1883(16.6%) 542(4.8%) 17(0.1%) 8081(71.2%) 295(2.6%) |
0.008 |
|
|
|
|
|
|
Surgery record Yes No Unknown |
2419(21.3%) 8854(78.0%) 76(0.7%) |
2655(23.4%) 8585(75.6%) 109(1.0%) |
<0.001 |
|
|
|
|
|
|
Radiation sequence No radiation and/or surgery Radiation after surgery Radiation prior to surgery Radiation before and after S Intraoperative radiation Sequence unknown, but both given Surgery before and after radiation Radiation in and before/after surgery |
9951(87.7%) 1218(10.7%) 156(1.4%) 12(0.1%) 3(0.0%) 14(0.1%) 0(0.0%) 0(0.0%) |
10076(88.8%) 1070(9.4%) 163(1.4%) 23(0.2%) 4(0.0%) 8(0.1%) 3(0.0%) 2(0.0%) |
0.008 |
|
|
|
|
|
|
Radiation record Beam radiation Beam with implants or isotopes Implant or radioisotopes Radiation, but not specified No radiation Unknown |
4680(41.2%) 8(0.1%) 18(0.2%) 58(0.5%) 6445(56.8%) 140(1.2%) |
4727(41.7%) 19(0.2%) 19(0.2%) 79(0.7%) 6399(56.4%) 106(0.9%) |
0.025 |
|
|
|
|
|
|
Chemotherapy record Yes No/unknown |
5408(47.7%) 5941(52.3%) |
5293(46.6%) 6056(53.4%) |
0.126 |
|
|
|
|
|
|
Tumor Size ≤1cm >1, ≤2cm >2, ≤3cm >3, ≤4cm >4 cm Unknown |
8567(75.5%) 7(0.1%) 9(0.1%) 9(0.1%) 6(0.1%) 2751(24.2%) |
8667(76.4%) 4(0.0%) 12(0.1%) 6(0.1%) 3(0.0%) 2657(23.4%) |
0.408 |
|
|
|
|
|
|
Bone Metastasis Yes No Unknown |
759(6.7%) 2937(25.9%) 7653(67.4%) |
613(5.4%) 3829(33.7%) 6907(60.9%) |
<0.001 |
|
|
|
|
|
|
Brain Metastasis Yes No Unknown |
668(5.9%) 3024(26.6%) 7657(67.5%) |
390(3.4%) 4034(35.5%) 6925(61.0%) |
<0.001 |
|
|
|
|
|
|
Liver Metastasis Yes No Unknown |
610(5.4%) 3086(27.2%) 7653(67.4%) |
374(3.3%) 4064(35.8%) 6911(60.9%) |
<0.001 |
|
|
|
|
|
|
Lung Metastasis Yes No Unknown |
551(4.9%) 3125(27.5%) 7673(67.6%) |
557(4.9%) 3896(34.3%) 6896(60.8%) |
<0.001 |
|
|
|
|
|
|
First malignant primary indicator Yes No |
9213(81.2%) 2136(18.8%) |
8807(77.6%) 2542(22.4%) |
<0.001 |
|
|
|
|
|
|
Age at diagnosis <65 ≥65 |
4300(37.9%) 7049(62.1%) |
3819(33.7%) 7530(66.3%) |
<0.001 |
|
|
|
|
|
|
Insurance status Any Medicaid Insured or no specifics Uninsured Unknown |
993(8.7%) 5655(49.8%) 4441(39.1%) |
1052(9.3%) 6368(56.1%) 240(2.1%) 3689(32.5%) |
<0.001 |
|
|
|
|
|
|
Marital status Married or domestic partner Divorced/separated/single/widowed Unknown |
5975(52.6%) 4995(44.0%) 379(3.3%) |
5922(52.2%) 4993(44.0%) 434(3.8%) |
0.138 |
|
|
|
|
|
|
High school education (%) ≤10 >10, ≤20 >20, ≤30 >30 Unknown |
2050(18.1%) 5984(52.7%) 2981(26.3%) 332(2.9%) 2(0.0%) |
2202(19.4%) 5822(51.3%) 2929(25.8%) 395(3.5%) 1(0.0%) |
0.008 |
|
|
|
|
|
|
Median family income (dollar, in tens) ≤5000 >5000, ≤7000 >7000, ≤9000 >9000 Unknown |
1682(14.8%) 5745(50.6%) 2854(25.1%) 1066(9.4%) 2(0.0%) |
1682(14.8%) 5470(48.2%) 2909(25.6%) 1287(11.3%) 1(0.0%) |
<0.001 |
|
|
|
|
|
Abbreviations: NSCLC: non-small cell lung cancer; LCC: large cell carcinoma. The bolded p-value of less than 0.05 represents a significant statistical difference.
Point 2. It is not clear from the tables that education and family income variables are at census track level. Please make this clear in the method section.
Response 2: Thank you very much to your reminding. This is a very constructive suggestion to improve the quality of the research. We added relevant expression in the method section (please see page 5, line 132 in modified version). Thanks again for your suggestions.
Point 3. Please add the information of the test used to generate the bolded p-values from the multivariate analysis in the method section.
Response 3: Thank you very much to your reminding. This is a very constructive suggestion to improve the quality of the article. We added relevant explanation in the method section (please see page 5, line 144 in the modified version). And we also added relevant captions of all the tables in the modified version (please see Table 1, 3, 4, 5, S1 in the modified version). Thanks again for your suggestions.
Point 4. Although the title indicated that radiation sequence is the focus, the study did not focus on that. For instance, in the discussion section on page 18, “In this study, we mainly explored the impact of clinical characteristics and the therapeutic strategies on the survival outcomes in LCC patients, especially the elderly patients.” This did not mention anything about radiation sequence. I think the title should be changed to reflect what this article is really about.
Response 4: Thank you very much for your constructive suggestions. Your suggestion is very to the point. The title was replaced by “The clinical characteristics and treatments for large cell carcinoma patients older than 65 years old: A population-based study” (please see page 1, line 1 in modified version). Thanks again for your suggestions.
Point 5. The sample sizes in radiation sequence categories other than before and after surgery are too small to make valid inferences about.
Response 5: Thank you for your review and valuable suggestions. Your suggestion is very to the point. Except for radiation before and after surgery, other sequences with surgery were so rare in treatments for LCC patients that SEER database only collected a small number of samples with these radiation sequences categories. However, radiotherapy before and after surgery were more common in LCC patients, so the comparison of the two radiation sequences with surgery has been debated for a long time in many literatures, such as literature 1 & 2. Our article was also focused on the survival benefits of the preoperative and postoperative radiotherapy. Thanks again for your suggestions.
[1]. Van Houtte, P.; Moretti, L.; Charlier, F.; Roelandts, M.; Van Gestel, D. Preoperative and postoperative radiotherapy (RT) for non-small cell lung cancer: still an open question. Translational lung cancer research 2021, 10, 1950-1959, doi:10.21037/tlcr-20-472.
[2]. Le Pechoux, C.; Pourel, N.; Barlesi, F.; Lerouge, D.; Antoni, D.; Lamezec, B.; Nestle, U.; Boisselier, P.; Dansin, E.; Paumier, A.; et al. Postoperative radiotherapy versus no postoperative radiotherapy in patients with completely resected non-small-cell lung cancer and proven mediastinal N2 involvement (Lung ART): an open-label, randomised, phase 3 trial. The Lancet. Oncology 2022, 23, 104-114, doi:10.1016/s1470-2045(21)00606-9.
Point 6. The table showing the distributions of the variables after matching should be included in the manuscript as a supplemental table at least.
Response 6: Thank you for your constructive suggestions. This is a very meaningful suggestion to improve the quality of the article. We added Table S1 in supplements (please see Table S1 in supplements). Thanks again for your suggestions.
Table S1. Baseline characteristics of patients with LCC and other types of NSCLC after PSM
|
Characteristics
|
LCC (n=11349) |
Others (n=11349) |
P |
|
|
|
|
|
|
Race White Black Asian and others Unknown |
9243(81.4%) 1568(13.8%) 528(4.7%) 10(0.1%) |
9312(82.1%) 1437(12.7%) 582(5.1%) 18(0.2%) |
0.012 |
|
|
|
|
|
|
Sex Male Female |
6618(58.3%) 4731(41.7%) |
6720(59.2%) 4629(40.8%) |
0.169 |
|
|
|
|
|
|
Year of diagnosis 2004-2007 2008-2011 2012-2015 |
5487(48.3%) 3508(30.9%) 2354(20.7%) |
4624(40.7%) 3764(33.2%) 2961(26.1%) |
<0.001 |
|
|
|
|
|
|
Region East Northern Plains Southwest Alaska and Pacific Coast |
5979(52.7%) 1237(10.9%) 343(3.0%) 3790(33.4%) |
5729(50.5%) 1346(11.9%) 313(2.8%) 3961(34.9%) |
0.002 |
|
|
|
|
|
|
Tumor location Upper lobe Middle lobe Lower lobe NOS Overlapping lesion Main bronchus Trachea |
5857(51.6%) 465(4.1%) 2573(22.7%) 1761(15.5%) 137(1.2%) 546(4.8%) 10(0.1%) |
5729(50.5%) 413(3.6%) 3147(27.7%) 1260(11.1%) 144(1.3%) 612(5.4%) 23(0.2%) |
<0.001 |
|
|
|
|
|
|
Grade Grade I Grade II Grade III Grade IV Unknown |
28(0.2%) 95(0.8%) 3368(29.7%) 2781(24.5%) 5077(44.7%) |
364(3.2%) 1862(16.4%) 2731(24.1%) 106(0.9%) 6286(55.4%) |
<0.001 |
|
|
|
|
|
|
Stage Stage I Stage II Stage III Stage IV Unknown |
1269(11.2%) 668(5.9%) 2972(26.2%) 5735(50.5%) 705(6.2%) |
1501(13.2%) 810(7.1%) 3572(31.5%) 4423(39.0%) 1043(9.2%) |
<0.001 |
|
|
|
|
|
|
Laterality Right-origin of primary Left - origin of primary Bilateral, single primary Paired, but no laterality Others |
6282(55.4%) 4420(38.9%) 172(1.5%) 401(3.5%) 74(0.7%) |
6162(54.3%) 4698(41.4%) 142(1.3%) 268(2.4%) 79(0.7%) |
<0.001 |
|
|
|
|
|
|
Lymphadenectomy 0-3 LNRs ≥4 LNRs Biopsy or aspiration Sentinel biopsy None Unknown |
585(5.2%) 1683(14.8%) 542(4.8%) 14(0.1%) 8216(72.4%) 309(2.7%) |
531(4.7%) 1883(16.6%) 542(4.8%) 17(0.1%) 8081(71.2%) 295(2.6%) |
0.008 |
|
|
|
|
|
|
Surgery record Yes No Unknown |
2419(21.3%) 8854(78.0%) 76(0.7%) |
2655(23.4%) 8585(75.6%) 109(1.0%) |
<0.001 |
|
|
|
|
|
|
Radiation sequence No radiation and/or surgery Radiation after surgery Radiation prior to surgery Radiation before and after S Intraoperative radiation Sequence unknown, but both given Surgery before and after radiation Radiation in and before/after surgery |
9951(87.7%) 1218(10.7%) 156(1.4%) 12(0.1%) 3(0.0%) 14(0.1%) 0(0.0%) 0(0.0%) |
10076(88.8%) 1070(9.4%) 163(1.4%) 23(0.2%) 4(0.0%) 8(0.1%) 3(0.0%) 2(0.0%) |
0.008 |
|
|
|
|
|
|
Radiation record Beam radiation Beam with implants or isotopes Implant or radioisotopes Radiation, but not specified No radiation Unknown |
4680(41.2%) 8(0.1%) 18(0.2%) 58(0.5%) 6445(56.8%) 140(1.2%) |
4727(41.7%) 19(0.2%) 19(0.2%) 79(0.7%) 6399(56.4%) 106(0.9%) |
0.025 |
|
|
|
|
|
|
Chemotherapy record Yes No/unknown |
5408(47.7%) 5941(52.3%) |
5293(46.6%) 6056(53.4%) |
0.126 |
|
|
|
|
|
|
Tumor Size ≤1cm >1, ≤2cm >2, ≤3cm >3, ≤4cm >4 cm Unknown |
8567(75.5%) 7(0.1%) 9(0.1%) 9(0.1%) 6(0.1%) 2751(24.2%) |
8667(76.4%) 4(0.0%) 12(0.1%) 6(0.1%) 3(0.0%) 2657(23.4%) |
0.408 |
|
|
|
|
|
|
Bone Metastasis Yes No Unknown |
759(6.7%) 2937(25.9%) 7653(67.4%) |
613(5.4%) 3829(33.7%) 6907(60.9%) |
<0.001 |
|
|
|
|
|
|
Brain Metastasis Yes No Unknown |
668(5.9%) 3024(26.6%) 7657(67.5%) |
390(3.4%) 4034(35.5%) 6925(61.0%) |
<0.001 |
|
|
|
|
|
|
Liver Metastasis Yes No Unknown |
610(5.4%) 3086(27.2%) 7653(67.4%) |
374(3.3%) 4064(35.8%) 6911(60.9%) |
<0.001 |
|
|
|
|
|
|
Lung Metastasis Yes No Unknown |
551(4.9%) 3125(27.5%) 7673(67.6%) |
557(4.9%) 3896(34.3%) 6896(60.8%) |
<0.001 |
|
|
|
|
|
|
First malignant primary indicator Yes No |
9213(81.2%) 2136(18.8%) |
8807(77.6%) 2542(22.4%) |
<0.001 |
|
|
|
|
|
|
Age at diagnosis <65 ≥65 |
4300(37.9%) 7049(62.1%) |
3819(33.7%) 7530(66.3%) |
<0.001 |
|
|
|
|
|
|
Insurance status Any Medicaid Insured or no specifics Uninsured Unknown |
993(8.7%) 5655(49.8%) 4441(39.1%) |
1052(9.3%) 6368(56.1%) 240(2.1%) 3689(32.5%) |
<0.001 |
|
|
|
|
|
|
Marital status Married or domestic partner Divorced/separated/single/widowed Unknown |
5975(52.6%) 4995(44.0%) 379(3.3%) |
5922(52.2%) 4993(44.0%) 434(3.8%) |
0.138 |
|
|
|
|
|
|
High school education (%) ≤10 >10, ≤20 >20, ≤30 >30 Unknown |
2050(18.1%) 5984(52.7%) 2981(26.3%) 332(2.9%) 2(0.0%) |
2202(19.4%) 5822(51.3%) 2929(25.8%) 395(3.5%) 1(0.0%) |
0.008 |
|
|
|
|
|
|
Median family income (dollar, in tens) ≤5000 >5000, ≤7000 >7000, ≤9000 >9000 Unknown |
1682(14.8%) 5745(50.6%) 2854(25.1%) 1066(9.4%) 2(0.0%) |
1682(14.8%) 5470(48.2%) 2909(25.6%) 1287(11.3%) 1(0.0%) |
<0.001 |
|
|
|
|
|
Abbreviations: NSCLC: non-small cell lung cancer; LCC: large cell carcinoma. The bolded p-value of less than 0.05 represents a significant statistical difference.
Please see the attachment.
